

# Coupled machine learning and the limits of acceptability approach applied in parameter identification for a distributed hydrological model

Aynom T. Teweldebrhan[1], John F. Burkhart[1], Thomas V. Schuler[1], Morten Hjorth-Jensen[1,2]

[1]University of Oslo, Oslo, Norway
[2]Michigan State University, USA

*Correspondence to*: Aynom T. Teweldebrhan (aynomtt@geo.uio.no)

**Abstract.** Monte Carlo (MC) methods have been widely used in uncertainty analysis and parameter identification for hydrological models. The main challenge with these approaches is, however, the prohibitive number of model runs required to get an adequate sample size which may take from days to months especially when the simulations are run in distributed mode. In the past, emulators have been used to minimize the computational burden of the MC simulation through direct estimation of the residual based response surfaces. Here, we apply emulators of MC simulation in parameter identification for a distributed conceptual hydrological model using two likelihood measures, i.e. the absolute bias of model predictions (Score) and another based on the time relaxed limits of acceptability concept (pLoA). Three machine learning models (MLMs) were built using model parameter sets and response surfaces with limited number of model realizations (4000). The developed MLMs were applied to predict pLoA and Score for a large set of model parameters (95000). The behavioural parameter sets were identified using a time relaxed limits of acceptability approach based on the predicted pLoA values; and applied to estimate the quantile streamflow predictions weighted by their respective Score. The three MLMs were able to adequately mimic the response surfaces directly estimated from MC simulations; and the models identified using the coupled ML emulators and the limits of acceptability approach have performed very well in reproducing the median streamflow prediction both during the calibration and validation periods.

## 1 Introduction

Conceptual hydrological models have wide range of applications in solving various water quantity and quality related problems. A conceptual model typically comprises one or more calibration parameters as part of the user's perception of the hydrological processes in the catchment and the corresponding simplifications that are assumed to be acceptable for the intended modelling purpose (Beven, 1989; Refsgaard et al., 1997). One of the major challenges in using conceptual models, however, is the identification of model parameters to a particular catchment (e.g. Bárdossy and Singh, 2008). The failure to set parameter values in accordance to their theoretical bounds, the interaction between these parameters, as well as the absence of a unique best set of parameters are some of the causes of parameter uncertainty (Abebe and Price, 2003; Renard et al., 2010). In light of the different sources of uncertainty, previous studies have pointed out the need for a rigorous uncertainty analysis and communicating model simulation results in terms of uncertainty bounds rather than with only crisp values (e.g. Uhlenbrook et al., 1999).

In the past, various uncertainty analysis techniques have been proposed to infer model parameter values from observations, including the generalized likelihood uncertainty estimation (GLUE) methodology (Beven and Binley, 1992), the dynamic identifiability analysis framework (DYNIA) (Wagener et al., 2003), the Shuffled Complex Evolution




Metropolis algorithm (SCEM) (Vrugt et al., 2003), and the Bayesian inference (Kuczera and Parent, 1998; Yang et al., 2007). The GLUE methodology was inspired by the generalized sensitivity analysis concept of Hornberger and Spear (1981) and it is the most widely used uncertainty analysis framework in hydrology (Stedinger et al., 2008; Xiong et al., 2008; Shen et al., 2012). The residual-based version of this framework allows the user to choose a likelihood and the threshold value for

identification of behavioural and non-behavioural models. The limits of acceptability based GLUE methodology (GLUE LoA) (Beven, 2006) overcomes limitations of the residual based GLUE, that arise from the subjectivity in choosing the likelihood and the threshold value, by setting error bounds around the observed values. Models whose prediction falling within the error bounds for all observations are considered behavioural. The original GLUE LoA, which was formulated as a rejectionist framework in testing environmental models as hypothesis, is too stringent to be used for calibration purpose

especially in continuous rainfall-runoff modelling. In the past, different approaches have been made to minimize the rejection of useful models when using GLUE LoA. These approaches include relaxing the limits (e.g. Blazkova and Beven, 2009; Liu et al., 2009), using different model realizations for different periods of a hydrological year (e.g., Choi and Beven, 2007) and using a time relaxed approach with the degree of relaxation constrained by an additional efficiency criterion (Teweldebrhan et al., 2018). The time relaxed GLUE LoA approach (hereafter referred as GLUE pLoA) was based on the

empirical relationship between model efficiency and uncertainty in response to the percentage of model predictions that fall within the observation error bounds (pLoA). In a case study involving this approach and an operational hydrological model, the ensemble of model realizations with only 30-40 % of their predictions in a hydrologic year falling within the observation error bounds were able to predict streamflow during the evaluation period with an acceptable degree of accuracy for the intended use based on the commonly used efficiency criteria.

Monte Carlo (MC) simulation is commonly employed to quantify the uncertainty propagated from model parameters to predictions in model calibration and uncertainty analysis frameworks including the GLUE methodology. MC simulation involves the sampling of very large parameter sets from their respective parameter dimension. This is especially true when the parameter distribution is not known a priori and hence a uniform distribution is assumed. Although, the MC simulation is a widely accepted stochastic modelling techniques, it suffers from heavy computational burden (Yu et al., 2015). The

computational time and resources required by the MC simulation could be prohibitively expensive in the case of computationally intensive models such as those with a distributed setup (e.g. Shrestha et al., 2014). In the past, different approaches have been introduced to minimize the computational burden by reducing the number of model realizations in MC simulation. These include the use of more efficient parameter sampling techniques such as the Latin hypercube sampling (e.g. McKay et al., 1979; Iman and Conover, 1980) and adaptive Markov chain MC sampling (e.g. Blasone et al., 2008; Vrugt et

al., 2009) as well as through use of emulators (e.g. Wang et al., 2015). An emulator or surrogate model is a computationally efficient model that is calibrated over a small dataset obtained by the simulation of a computationally demanding model and used in its place during computationally expensive tasks (Pianosi et al., 2016).

In hydrology, surrogate modelling has been mainly used in optimization and sensitivity analysis frameworks (Oakley and O'Hagan, 2004; Emmerich et al., 2006; Razavi et al., 2012). This approach involves a limited number of model realizations

to build a surrogate model using the parameter sets and model outputs as covariates and independent variable, respectively. Statistical (e.g. Jones, 2001; Hussain et al., 2002; Regis and Shoemaker, 2004), Gaussian processes (Kennedy and O'Hagan, 2001; Yang et al., 2018) and machine learning models (MLMs) (e.g. Yu et al., 2015) have been used as surrogate models to emulate MC simulations. A machine learning model estimates the dependency between the inputs and outputs of a system from the available data (Mitchell, 1977).



In this study three MLMs, i.e. random forest (RF), K-nearest neighbours (KNN), and artificial neural network (NNET) are built using limited number of model parameter sets and response surfaces to emulate the MC simulation through coupling with the limits of acceptability approach. In hydrology, machine learning approaches have been increasingly used in different areas of application following the improvement in computation power. MLMs have been used in direct

prediction of different water quantity variables such as streamflow (Solomatine and Shrestha, 2009; Modaresi et al., 2018; Senent-Aparicio et al., 2018), evapotranspiration (Torres et al., 2011) and snow water equivalent (Marofi et al.,2011; Buckingham et al., 2015; Bair et al., 2017). Similarly MLMs have been used to predict water quality related variables such as nitrate concentration (Ransom et al., 2017) and sediment transport (Bhattacharya et al., 2017). MLMs have also been used to forecast the residuals of a conceptual rainfall-runoff model (Abebe and Prince, 2003) and as emulator for conducting

parameter uncertainty analysis of a conceptual hydrological model in order to overcome the high computational cost of the MC simulation (Shrestha et al., 2009).

The main goal of this study is to emulate the time consuming MC simulation for parameter identification through coupling of the machine learning models with the time relaxed limits of acceptability approach. The first objective is to assess the possibility of using pLoA as a likelihood measure for identification of behavioural models using the coupled

MLMs and the limits of acceptability approach, instead of the previously used residual-based likelihood measures. The second objective is to compare the relative performances of RF and KNN as emulators of the MC simulation in relation to the standard machine learning based emulator, i.e. NNET. As of our best knowledge, RF and KNN have not been used before as emulators of the MC simulation in parameter identification for hydrological models. The third objective is to compare the performance of the MLMs trained using pLoA against those trained using the absolute bias based criterion

(Score) as target variables in conducting sensitivity analysis in order to assess the relative influence of the model parameters on the simulation result.

This paper is structured as follows: Section 2 presents a brief introduction to parameter identification using the time relaxed GLUE LoA approach as well as the MLMs used in this study. This section will also present the procedure followed in coupling the MLMs with the time relaxed GLUE LoA to emulate the MC simulation. Section 3 introduces the

hydrological model and the study area used in the case study. Section 4 presents the validation results of the ML models through comparison of the predicted response surfaces against those directly generated from the MC simulation as well as comparison of the simulated streamflow from behavioural models identified using the coupled MLMs and the time relaxed GLUE LoA against the observed values. Implications of the results in relation to the dataset and models used in this study as well as relevant previous studies are discussed in Section 5 and conclusions are drawn in section 6.

## 2  Methodology

Coupling of the MLMs with the GLUE pLoA was realized in two main phases. In the first phase, the response surfaces were generated using limited number of MC simulations with subsequent evaluation of each realization using pLoA and Score as likelihood measures. The MLMs were then built using the parameter sets and the response surfaces. In the second phase, the developed MLMs were applied to predict the response surfaces for new parameter sets and the GLUE pLoA was used to

identify the behavioural parameter sets based on the predicted response surfaces. The R software and its package for classification and regression training (CARET) were used for building and application of the MLMs as well as for conducting further analyses.



### 2.1 Parameter identification using the time-relaxed limits of acceptability approach

The GLUE methodology (Beven and Binley, 1992) accepts the condition in which different behavioural model realizations give comparable model results, i.e. equifinality, as a working paradigm for parameter identification of hydrological models (Choi and Beven, 2007). The first step followed in implementing this methodology was identification of the uncertain model

parameters and setting the range of their respective dimensions. The next step was to randomly sample the parameter sets from the prior distribution. Since the parameter distribution was not known a priori, a uniform MC sampling was employed. The hydrological model was run using the sampled parameter sets and the streamflow predictions of all model realizations were retrieved for further analysis.

       The GLUE limits of acceptability approach (GLUE LoA) (Beven, 2006) was used to characterize behavioural and non-

behavioural simulations. This approach relies on an assessment of uncertainty in the observational data and involves setting an observation error bounds (limits) with due consideration to the observation and other sources of uncertainties such as from the input data. Since no streamflow uncertainty data were available in the study site, mean observational uncertainty of 25% was assumed and the streamflow limits were defined using this value. In this study, the time relaxed variant of the GLUE LoA (GLUE pLoA) was employed to characterize behavioural models. In GLUE pLoA, the requirement in the

original formulation for the model realizations to satisfy the limits in 100% of the observations is relaxed; with the degree of relaxation constrained as a function of an acceptable modelling uncertainty expressed by the containing ratio index ($CR$). In previous studies involving the GLUE methodology, this index has been used as estimate of the prediction uncertainty (e.g. Xiong et al., 2009) and it is expressed as the number of observations falling within their respective prediction bounds to the total number of observation (Eq. 1).

$$CR = \frac{\sum_{i=1}^{n} I(Q_{obs,i})}{n} \qquad (1)$$

$$\text{with, } I(Q_{obs,i}) = \begin{cases} 1, & L_{lim,i} < Q_{obs,i} < U_{lim,i} \\ 0, & Otherwise \end{cases}$$

where $Q_{obs,i}$ represents observed streamflow at the the $i^{th}$ time step, and $L_{lim,i}$ and $U_{lim,i}$ respectively denote the lower and upper prediction bounds.

       The procedure followed in GLUE pLoA for relaxing the original formulation is detailed in Teweldebrhan et al. (2018). For completeness, we include a summary of the steps herein:

**Step 1:** define an acceptable modelling uncertainty (CR) at a chosen certainty level (e.g. 5-95 %). In this study the CR value

25        obtained for the calibration period using the residual based GLUE methodology was adopted as an acceptable CR value.

**Step 2:** relax the acceptable percentage of observations where model predictions fall within the limits. This is done by gradually lowering the requirement for bracketing the observations in 100% of the time steps up to the acceptable pLOA.

**Step 3:** test whether each model realization prediction falls within the limits at least for the specified percentage of the total observations. If model realizations that satisfy the relaxed acceptability criteria are found, proceed to step 4, otherwise

30        lower the threshold pLOA further and repeat this step.

**Step 4:** calculate the new CR and check if it is close to the predefined acceptable CR value. If the calculated CR is less than the predefined CR, repeat steps 2 to 4. Whereas, if the two CR values are close (e.g. within 5%) then accept all model realizations that satisfy this pLOA as behavioral and store their indices for use in further analysis.

The percentage of observations where model predictions fall within the limits, i.e. pLoA is estimated using Equation 2.



$$pLoA = \frac{\sum_{i=1}^{n} S(Q_{sim,i})}{n} * 100 \tag{2}$$

$$\text{with, } S(Q_{sim,i}) = \begin{cases} 1, & L_{e,i} < Q_{sim,i} < U_{e,i} \\ 0, & Otherwise \end{cases}$$

where $Q_{sim,i}$ represents simulated streamflow corresponding to the i[th] observation, and $L_{e,i}$ and $U_{e,i}$ are the lower and upper observation error bounds, respectively.

The identified behavioural model realizations were used to predict streamflow weighted by their respective Score values. When calculating Score, the prediction error, i.e. the deviation between the observed and simulated streamflow ($Q$) values was first converted into a normalized criterion. This was accomplished using a triangular membership function with its support representing the uncertainty in streamflow observations and the pointed core representing a perfect match between the observed and predicted values (Eq. 3). Following that, the total Score ($S_j$) of each model realization, $j$, was calculated as the membership grade of the prediction error, summed over all observations (Eq. 4) and the normalized weight in relation to the other model realizations ($w_j$) was calculated using Eq. 5.

$$\mu_Q(e) = \begin{cases} 0, e \leq L_e \\ \dfrac{e - l}{m - l}, L_e < e \leq m \\ \dfrac{u - e}{u - m}, m < e < U_e \\ 0, e \geq U_e \end{cases} \tag{3}$$

$$S_j = \sum_{i=1}^{n} \mu_Q(e) \tag{4}$$

$$w_j = \frac{S_j}{\sum_{k=1}^{N} S_k} \tag{5}$$

where $\mu_Q(e)$ is the membership grade of each prediction error ($e$) corresponding to the observed streamflow value $i$; $m$ is the point in the support with perfect match between the observed and predicted streamflow values. The variables $L_e$ and $U_e$ respectively refer to the lower and upper error bounds of the streamflow observations. The number of streamflow observations and behavioural models are respectively denoted by n and N.

### 2.2 Machine learning modelling

Three non-linear and non-parametric machine learning methods, i.e. RF, KNN, and NNET from the CARET package of R (Kuhn, 2008) were considered to emulate the MC simulation. In all methods, relevant parameters were optimized and the root mean squared error (RMSE) metric was used to identify the optimal model. This section briefly summarizes these machine learning methods and the reader is referred to the above reference for detailed description of these algorithms.

### 2.2.1 Random forest

Random forest (RF) is a version of the Bagged (bootstrap-aggregated) trees algorithm (Breiman, 2001). As such, it is an ensemble method whereby a large number of individual trees are grown from random subsets of predictors, providing a weighted ensemble of trees (Bair et al. 2017). Bagging was reported to be a successful approach for combining unstable





learners (e.g. Li et al., 2011). Since RF combines bagging with a randomization of the predictor variables used at each node, it yields an ensemble of low correlation trees (Li et al., 2011, Appelhans et al., 2015). The free parameter in this method, i.e. the number of randomly selected predictors at each node, was determined through optimization. RF is also less sensitive to non-important variables, since it implicitly performs variable selection (Okun and Priisalu, 2007).

### 2.2.2 K-nearest neighbors

K-nearest neighbors (KNN) approach uses the K-closest samples from the training dataset to predict a new sample. The value of K, i.e. the number of closest samples used in the final model was optimized. KNN is a nonparametric method where the information extracted from the observed datasets is used to predict the variable of interest without defining a predetermined parametric relationship between the predictors and predicted variables (Modaresi et al., 2018). KNN is also a non-linear method whose prediction solely depends on the distance of the predictor variables to the closest training dataset known to the model (Appelhans et al., 2015). In this study, the Euclidean distance was used as a similarity measure for computing the distance between the new and training datasets.

### 2.2.3 Artificial neural network

An artificial neural network (NNET) constitutes an interconnected and weighted network of several simple processing units called neurons that are analogous to the biological neurons of the human brain (Hsieh, 1993; Tabari et al., 2010). The neurons provide the link between the predictors and the predicted variable and in the case of supervised learning the weights of the neurons, i.e. the unidirectional connection strengths, are iteratively adjusted to minimize the error (Sajikumar and Thandaveswara, 1999; Bair et al. 2018). NNET has the capability to detect and learn complex and nonlinear relationships between variables (Yu et al., 2015).

A multilayer perceptron is the most common type of neural network used in supervised learning (Zhao et al., 2005; Marofi et al., 2011) and it consists of an input layer in which input data are fed, one or more hidden layers of neurons in which data are processed, and an output layer that produces output data for the given input (e.g. Senent-Aparicio et al., 2018). In this study one middle layer was considered, with the number of neurons in the input and output layers being equal to the number of predictors and predicted variable, respectively. The two free parameters of NNET, i.e. the number of neurons in the middle layer and the value of weight decay were optimized. Based on a preliminary assessment on performances of models with a linear and sigmoid activation function, a linear activation function was used in the final model.

### 2.3 Coupling of the machine learning emulators with the limits of acceptability approach

The procedure followed to build and apply the MLMs as emulators of the MC simulation is similar to the approach used in previous studies (e.g. Yu et al., 2015) with the exception of the parameter identification part. While the previous studies were conducted based on the residual based GLUE, here we use the time relaxed GLUE LoA approach with two likelihood measures. The coupling procedure involved sampling of 5000 random samples from the dimensions of the uncertain model parameters. The hydrological model was run using these parameter sets with subsequent retrieval of the simulated streamflow values. Each model realization was evaluated both in terms of its capability to generate simulated streamflow close to the observed values (Score) and its persistency in producing acceptable simulated values that fall within the observation error bounds (pLoA). Six MLMs (for the combinations of the two likelihoods, i.e. Score and pLoA and for the three ML methods, i.e. RF, KNN, and NNET) were trained and tested using the randomly selected parameter sets and their





corresponding likelihood values directly estimated from the MC simulation. Sample sizes of 80% (S1) and 20% (S2) of the 5000 samples were respectively used for training and testing the MLMs (Table 1).

**Table 1.** Parameter samples used in building and application of the MLM-based emulators.

| Sample | Size | Description |
|--------|------|-------------|
| S1 | 4000 | used for training the MLMs |
| S2 | 1000 | used for testing the MLMs |
| S3 | 95000 | used to predict the response surface |
| S4 | - | identified behavioural samples |

The trained MLMs were applied to emulate the MC simulation through prediction of the likelihood measures
corresponding to a much bigger size of randomly generated parameter sets, i.e. 95000 (S3). For further validation of the MLMs, an MC simulation was also run using the hydrological model and the S3 parameter sets with subsequent retrieval of the simulated streamflow and estimation of the two likelihood measures through comparison of the simulated against observed streamflow values. Performance of the surrogate models to emulate the MC simulation was evaluated through comparison of their likelihood prediction against those estimated from the MC simulation. The identification of behavioural
parameter sets (S4) from the S3 samples was realized using the time relaxed GLUE LoA approach based on the MLM predicted pLoA values of the samples. The score-weighted streamflow predictions of these behavioural models were calculated at different quantile values. Performance of the three MLMs as emulators of the MC simulation was further assessed through cross-validation of the streamflow predictions of behavioural models identified using each MLM coupled with GLUE pLoA (MLM-GLUE pLoA) against observed values in the remaining years other than the one used for building
the MLM-GLUE pLoA.

The procedure followed in building and evaluation of MLM-GLUE pLoA can be divided into two main phases as outlined below and depicted as schematic overview in Fig. 1:

(a) MLM training and testing

    i.    Randomly sample 5000 parameter sets from their respective parameter dimensions.

ii.    Run the hydrological model using the sampled parameter sets and store the simulated streamflow corresponding to each parameter set.

   iii.   Calculate the performance of each model realization in terms of the percentage of time steps it is able to reproduce the observed streamflow within the observation error bounds, i.e. pLoA, and the total normalized absolute bias of the predicted streamflow (Score). A streamflow observation error bound of 25% was assumed in this study.

iv.   Use 80% of the parameter sets, i.e. S1, of the samples generated at step i as covariates; and the performance of each parameter set (pLoA) calculated at the previous step as target variable to train the MLMs i.e. RF, KNN, and NNET (MLMs_pLoA). Similarly, train the three MLMs using same parameter sets (S1) as covariates but with Score as a target variable (MLMs_score).

   v.    Test the trained MLMs_pLoA using the remaining 20% of the parameter sets generated at step i, i.e. S2, and the
30       corresponding target variable (pLoA) from step iii. Similarly, test the trained MLMs_score using the same samples (S2) but with Score as a target variable.

(b) Response surface estimation and behavioural model identification

    i.    Repeat the steps i to iii in MLM training and testing (a) but with a much bigger sample size of 95000 (S3)

   ii.    Predict pLoA and Score using MLMs_pLoA and MLMs_score, respectively and S3 as covariate.





iii.   Identify behavioural samples (S4) from S3 using the time relaxed limits of acceptability approach (Section 2.1) based on the pLoA predicted by the MLM.

iv.   Estimate weighted median streamflow prediction of the behavioral models. The Score predicted by the MLMs_score was first normalized using Eq. 5 and then used to weigh the relative contribution of each model realization.

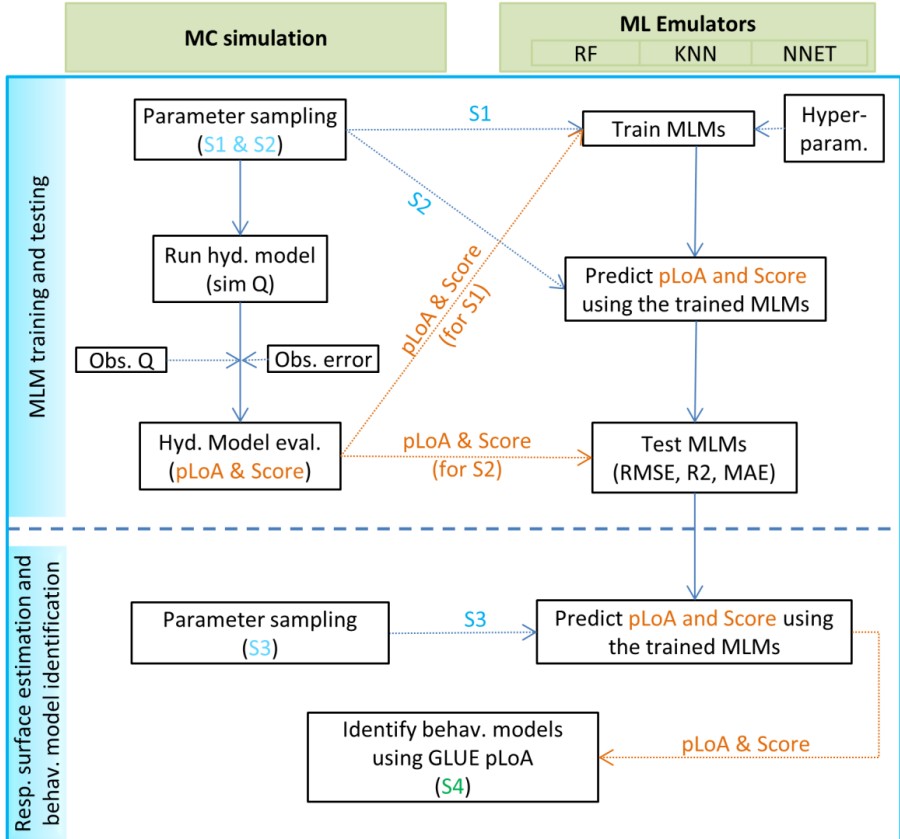

**Figure 1**. Schematic overview of the MLM training and testing as well as response surface prediction using the MLMs and the identification of behavioural models using the coupled MLM and GLUE pLoA.

**2.4 Model performance measures**

The performances of the generated ML models, i.e. RF, KNN, and NNET in terms of their capability to reproduce the response surfaces were evaluated using the following three standard statistical criteria, i.e. root mean square error ($RMSE$), coefficient of determination ($R^2$) and the mean absolute bias ($MAB$).

$$RMSE = \sqrt{\frac{1}{N} \sum_{i=1}^{N} \left( L_{ml,i} - L_{mc,i} \right)^2} \qquad (6)$$


$$R^2 = \frac{\left[\sum_{i=1}^{N}\left(L_{mc,i} - \bar{L}_{mc}\right)\left(L_{ml,i} - \bar{L}_{ml}\right)\right]^2}{\sum_{i=1}^{N}\left(L_{mc,i} - \bar{L}_{mc}\right)^2 \cdot \sum_{i=1}^{N}\left(L_{ml,i} - \bar{L}_{ml}\right)^2} \tag{7}$$

$$MAB = \frac{1}{N}\sum_{i=1}^{N}\left| L_{ml,i} - L_{mc,i}\right| \tag{8}$$

where $L_{ml,i}$ and $L_{mc,i}$ respectively denote the likelihood values (pLoA or Score) predicted using a given MLM and estimated using the MC simulation for the $i^{th}$ model realization. $\bar{L}_{ml}$ and $\bar{L}_{mc}$ are the average MLM predicted and MC estimated likelihood values, respectively. N is the total number of model realizations.

The Nash-Sutcliffe efficiency (NSE, Eq. 9) and the NSE with log-transformed data (LnNSE) were used for assessing the
streamflow prediction of behavioral models identified using MLM-GLUE pLoA through comparison against the observed values.

$$NSE = 1 - \frac{\sum_{i=1}^{n}(Q_{sim,i} - Q_{obs,i})^2}{\sum_{i=1}^{n}(Q_{obs,i} - \bar{Q}_{obs})^2} \tag{9}$$

where $Q_{sim,i}$ and $Q_{obs,i}$ respectively represent simulated and observed streamflow for the $i^{th}$ time step and $\bar{Q}_{obs}$ represents mean value of the observed streamflow series.

## 3    Case study

### 3.1 The hydrological model

The Statkraft Hydrological Forecasting Toolbox, Shyft, (https://github.com/statkraft/shyft) is an open-source distributed hydrological modelling framework developed by Statkraft (2018) with contributions from the University of Oslo and other institutions (e.g. Nyhus, 2017; Matt et al., 2018). The modelling framework has three main models (method stacks) and in this study, the PT_GS_K model was used for the parameter identification study using machine learning based emulators of
the MC simulation. PT_GS_K is a conceptual hydrological model and in this study eight of its parameters are subjected to uncertainty analysis. PT_GS_K uses the Priestley-Taylor (PT) method (Priestley and Taylor, 1972) for estimating potential evaporation; a quasi-physical based method for snow melt, sub-grid snow distribution and mass balance calculations (GS method); and a simple storage-discharge function (Lambert, 1972; Kirchner, 2009) for catchment response calculation (K). Overall, these three methods constitute the PT_GS_K model in Shyft. The framework establishes a sequence of spatially
distributed cells of arbitrary size and shape. As such it can provide lumped (single cell) or discretized (spatially distributed) calculations, as in this study. The modelling framework (shyft) and the PT_GS_K model in particular were described in previous studies (e.g. Burkhart et al., 2016; Teweldebrhan et al., 2018) and the reader is referred to these materials for further details on the underlying methods of this model. The following table shows list of the uncertain model parameters and their parameter range.



**Table 2.** Range of model parameters used for the PT_GS_K model uncertainty analysis

| Model Parameter | Min. | Max. | Description |
|---|---|---|---|
| c1 | -5.0 | 1.0 | constant in Catchment Response Function, CRF |
| c2 | 0.0 | 1.2 | linear coefficient in CRF |
| c3 | -0.15 | -0.05 | quadratic coefficient in CRF |
| tx | -3.0 | 2.0 | Solid/liquid threshold temperature ($^o$C) |
| ws | 1.0 | 6.0 | wind scale, i.e. slope in turbulent wind function |
| fa | 1.0 | 15.0 | fast albedo decay rate (days) |
| sa | 20.0 | 40.0 | slow albedo decay rate (days) |
| cv | 0.06 | 0.85 | spatial coefficient of variation of snowfall |

### 3.2 Study site and data

The data used for training and validation of the ML emulators was retrieved from the Nea-catchment. This catchment is located in Sør-Trøndelag County, Norway (Fig. 2). Geographical location of the catchment ranges from 11.67390 $^o$ to
12.46273 $^o$ E and from 62.77916 $^o$ to 63.20405 $^o$ N. The Nea-catchment covers a total area of 703 km$^2$ and it is characterized by a wide range of physiographic and land cover characteristics. Altitude of the catchment ranges from 1783 masl on the eastern part around the mountains of Storsylen to 649 masl at its outlet. The dominant land cover types in the catchment are moors, bogs, and some sparse vegetation, while limited part of the catchment is forest covered (3%). Mean annual precipitation for the hydrological years 2011-2014 was 1120 mm. The highest and lowest average daily temperature values
for this period were 28 $^o$C and -30 $^o$C, respectively.

       PT_GS_K model requires temperature, precipitation, radiation, relative humidity, and wind speed as forcing data. In this study, daily time series data of these variables were obtained from Statkraft (2018) with the exception of relative humidity. Daily gridded relative humidity data was retrieved from ERA-interim (Dee et al., 2011). The model also requires the following physiographic data of each grid cell: average elevation, grid cell total area, and the areal fractions of forest,
reservoir, lake, and glacier. Data for these physiographic variables were retrieved from two sources: the land cover data from Copernicus land monitoring service (2016) and the 10m digital elevation model (10m DEM) from the Norwegian mapping authority (2016). Daily observed streamflow measurements that were used both in behavioral model identification and validation that cover four hydrological years (September 1 to August 31) for the study area were also provided by Statkraft (2018).


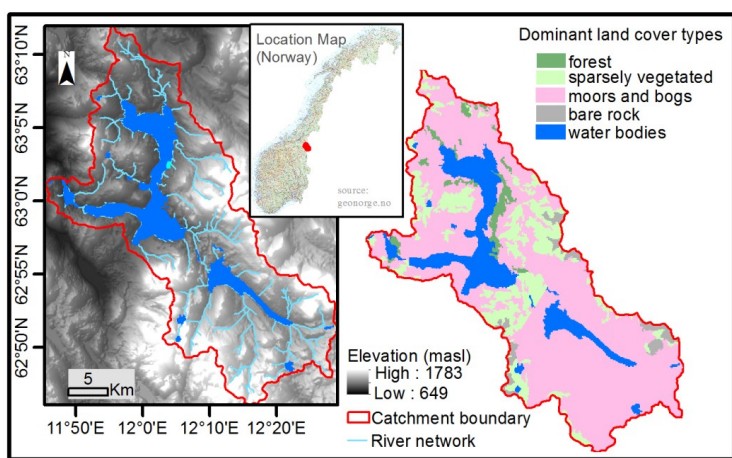

**Figure 2**. Physiography and location map of the study domain

## 4 Results

### 4.1 Evaluation of the MLMs capability in reproducing the response surfaces

Table 3 shows the test and validation results of the MLMs trained to emulate the MC simulation. Two sets of MLM emulators were trained using the same covariates (S2) but different target variables, i.e. pLoA and Score. The pLoA and Score predicted using the test and validation samples were compared against their respective values estimated using the MC simulation. The evaluation metrics have shown variability both between the three MLMs and the analysis years. For the test samples and using pLoA as a target variable, while similar results were obtained between RF and NNET, a relatively lower

performance was observed when using KNN. The highest performance of the MLMs was observed in year 2014 with $R^2$ value of 0.91, 0.76 and 0.92 for RF, KNN, and NNET respectively and the lowest performance was observed in year 2013 with $R^2$ value of 0.86, 0.7 and 0.85 for RF, KNN, and NNET respectively. When using Score as a target variable and the test samples, RF, NNET, and KNN have shown a decreasing order of performance based on the three evaluation metrics, i.e. RMSE, $R^2$, and MAE. The inter-annual comparison of the evaluation metrics shows that the relative performance of the

MLMs using Score as a target variable was consistent throughout the four analysis years. Relative performances similar to the test samples were obtained for the validation samples both for MLMs_pLoA and MLMs_score.

**Table 3.** Evaluation result of the predicted target variables, i.e. pLoA (in fraction) and Score through comparison against values estimated using the MC simulation for the test and validation samples.

| Year | Metrics | Test (pLoA) | | | Validation (pLoA) | | | Test (Score) | | | Validation (Score) | | |
|------|---------|------|------|------|------|------|------|------|------|------|------|------|------|
| | | RF | KNN | NNET | RF | KNN | NNET | RF | KNN | NNET | RF | KNN | NNET |
| 2011 | RMSE | 0.028 | 0.041 | 0.027 | 0.028 | 0.042 | 0.029 | 4.698 | 7.058 | 5.510 | 4.710 | 6.964 | 5.417 |
| | $R^2$ | 0.888 | 0.751 | 0.884 | 0.884 | 0.741 | 0.872 | 0.876 | 0.721 | 0.821 | 0.875 | 0.727 | 0.827 |
| | MAE | 0.016 | 0.027 | 0.019 | 0.016 | 0.028 | 0.019 | 2.604 | 4.691 | 3.254 | 2.751 | 4.632 | 3.219 |
| 2012 | RMSE | 0.034 | 0.048 | 0.032 | 0.034 | 0.047 | 0.031 | 5.656 | 7.500 | 6.892 | 6.093 | 8.313 | 7.564 |
| | $R^2$ | 0.867 | 0.734 | 0.876 | 0.858 | 0.734 | 0.880 | 0.856 | 0.754 | 0.780 | 0.852 | 0.725 | 0.763 |
| | MAE | 0.020 | 0.030 | 0.021 | 0.019 | 0.030 | 0.020 | 3.343 | 4.887 | 4.133 | 3.453 | 5.206 | 4.437 |
| 2013 | RMSE | 0.034 | 0.049 | 0.034 | 0.034 | 0.050 | 0.034 | 5.001 | 8.030 | 6.508 | 5.787 | 8.670 | 7.274 |
| | $R^2$ | 0.862 | 0.701 | 0.847 | 0.865 | 0.699 | 0.854 | 0.876 | 0.675 | 0.786 | 0.862 | 0.687 | 0.772 |
| | MAE | 0.017 | 0.031 | 0.021 | 0.017 | 0.031 | 0.021 | 2.843 | 5.196 | 4.250 | 3.032 | 5.375 | 4.531 |
| 2014 | RMSE | 0.023 | 0.038 | 0.022 | 0.024 | 0.040 | 0.022 | 4.274 | 7.010 | 4.354 | 4.303 | 7.027 | 4.493 |
| | $R^2$ | 0.914 | 0.764 | 0.919 | 0.916 | 0.764 | 0.923 | 0.908 | 0.753 | 0.900 | 0.908 | 0.755 | 0.895 |
| | MAE | 0.014 | 0.026 | 0.015 | 0.014 | 0.026 | 0.015 | 2.569 | 4.693 | 2.870 | 2.532 | 4.663 | 2.897 |

**4.2 Evaluation of behavioural parameter sets using observed streamflow**

The behavioural model realizations identified using the coupled ML emulators and the limits of acceptability approach were evaluated using observed streamflow values. A cross-validation method was used to assess the performance of the model parameter sets identified in a given year through comparison of the simulated against observed streamflow values in the remaining years. The cross-validation result based on the streamflow efficiency measures used in this study, i.e. NSE and LnNSE as well as the CR are depicted in Table 4. The behavioural model realizations have performed very well both during the calibration and validation periods. During the calibration period, minimum NSE of 0.81, 0.89, and 0.82 were respectively obtained for the models identified using RF, KNN, and NNET as emulators. Similarly, the maximum NSE values during this period were 0.93, 0.94, and 0.95 respectively for RF, KNN, and NNET. The average NSE for these emulators was 0.88, 0.91, and 0.88, respectively. During the validation period the value of NSE ranged 0.72-0.83, 0.66-0.85 and 0.71-0.83 respectively for RF, KNN, and NNET. A relatively lower Ln_NSE value than NSE was observed in most of the analysis years with the exception of year 2012, where a relatively higher Ln_NSE was obtained than NSE when using RF and NNET during the calibration period. While a slightly higher average NSE was obtained when using KNN as compared to RF and NNET both during calibration (0.91) and validation (0.85) periods, a slightly higher average LnNSE was obtained when using NNET both during calibration (0.85) and validation (0.79) periods.

The measure of streamflow prediction uncertainty used in this study, i.e. CR, for the validation period has shown some variability based on the MLM used in behavioural model identification. When using RF, the highest and lowest CR values obtained were 0.65 and 0.89, respectively, with an overall mean value of 0.74. Similarly, minimum, maximum, and mean CR values respectively of 0.64, 0.80, and 0.71 were obtained when using NNET. The validation period CR values when using KNN ranged from 0.72 to 0.89 with an average value of 0.79, which is relatively higher as compared to RF and NNET.

The inter-annual comparison between the three MLMs shows that the highest validation period average NSE (0.89) was obtained under the year 2014 as calibration period and KNN as ML emulator. Similarly, the highest average LnNSE (0.86) for the validation period was obtained when using models calibrated in year 2014 but NNET as ML emulator. On the other hand, the lowest average NSE (0.74) for the validation period was obtained when using year 2013 as calibration period and RF and KNN as ML emulators. This shows that models identified based on KNN were characterized by a relatively higher





inter-annual variability in their performances (based on NSE) as compared to those identified using RF and NNET. A relatively higher inter-annual variability in average CR (0.66 to 0.79) for the validation periods was obtained when using RF.

**Table 4.** Cross-validation of the streamflow predictions of models identified using the coupled ML emulators and MC simulation.

| Emul. (MLM) | Calib year | Validation year | | | | | | | | | | | |
| | | 2011 | | | 2012 | | | 2013 | | | 2014 | | |
| | | NSE | LnNSE | CR | NSE | LnNSE | CR | NSE | LnNSE | CR | NSE | LnNSE | CR |
| RF | 2011 | **0.81** | **0.75** | **0.76** | 0.73 | 0.73 | 0.87 | 0.93 | 0.90 | 0.80 | 0.87 | 0.70 | 0.69 |
| | 2012 | 0.87 | 0.80 | 0.66 | **0.88** | **0.91** | **0.83** | 0.87 | 0.84 | 0.65 | 0.85 | 0.68 | 0.66 |
| | 2013 | 0.73 | 0.68 | 0.77 | 0.72 | 0.55 | 0.89 | **0.93** | **0.93** | **0.83** | 0.77 | 0.56 | 0.71 |
| | 2014 | 0.84 | 0.76 | 0.69 | 0.80 | 0.79 | 0.78 | 0.93 | 0.83 | 0.70 | **0.91** | **0.72** | **0.66** |
| KNN | 2011 | **0.89** | **0.80** | **0.80** | 0.79 | 0.83 | 0.86 | 0.94 | 0.88 | 0.82 | 0.90 | 0.73 | 0.73 |
| | 2012 | 0.86 | 0.80 | 0.72 | **0.91** | **0.90** | **0.88** | 0.89 | 0.79 | 0.72 | 0.88 | 0.68 | 0.72 |
| | 2013 | 0.80 | 0.72 | 0.80 | 0.66 | 0.59 | 0.88 | **0.94** | **0.93** | **0.86** | 0.75 | 0.61 | 0.75 |
| | 2014 | 0.88 | 0.79 | 0.81 | 0.85 | 0.85 | 0.89 | 0.94 | 0.82 | 0.82 | **0.91** | **0.72** | **0.72** |
| NNET | 2011 | **0.88** | **0.82** | **0.68** | 0.80 | 0.86 | 0.80 | 0.92 | 0.91 | 0.76 | 0.88 | 0.73 | 0.66 |
| | 2012 | 0.85 | 0.83 | 0.68 | **0.88** | **0.92** | **0.83** | 0.86 | 0.87 | 0.71 | 0.84 | 0.69 | 0.64 |
| | 2013 | 0.82 | 0.72 | 0.71 | 0.71 | 0.63 | 0.76 | **0.95** | **0.95** | **0.82** | 0.78 | 0.60 | 0.68 |
| | 2014 | 0.87 | 0.82 | 0.67 | 0.74 | 0.84 | 0.70 | 0.90 | 0.92 | 0.76 | **0.82** | **0.72** | **0.60** |

Figure 3 shows the scatter plots of simulated against observed streamflow for a sample calibration period (year 2011) and validation periods (years 2012, 2013, and 2014). The streamflow predictions for the calibration period have shown good fit with the observed values with most of the predicted values falling close to the 1:1 identity line (dark line). However, the small patch of the scatter points between 50 and 75 (m$^3$/s) of the observed values show underestimation for this streamflow range. This might be attributed to poor estimation of the model parameters or due to an interaction of the model parameters that had a significant effect on dominating processes in that flow range. In years 2012 and 2014, the predicted streamflow

has shown good fit with the low-flow observations. A mismatch was observed with the high-flow observations during the same period, where most of the high-flow observations are underestimated. These years are characterized by having the highest (year 2012) and lowest (year 2014) maximum SWE (data not shown) as compared to the other years and this may partly explain to the observed low performance during the high-flow condition. The behavioural models identified using the three MLMs yielded very good streamflow prediction in year 2013. From the trend line fitted to the scatters, it can be

noticed that the predictions based on RF tend to slightly underestimate for high-flow conditions and overestimate for low-flow conditions in years 2012 and 2014 as compared to KNN and NNET. The latter MLMs yielded fitted lines close to each other in both the calibration and validation periods with the exception of year 2013, where KNN and NNET respectively yielded slightly over- and underestimated streamflow predictions for the high-flow condition.



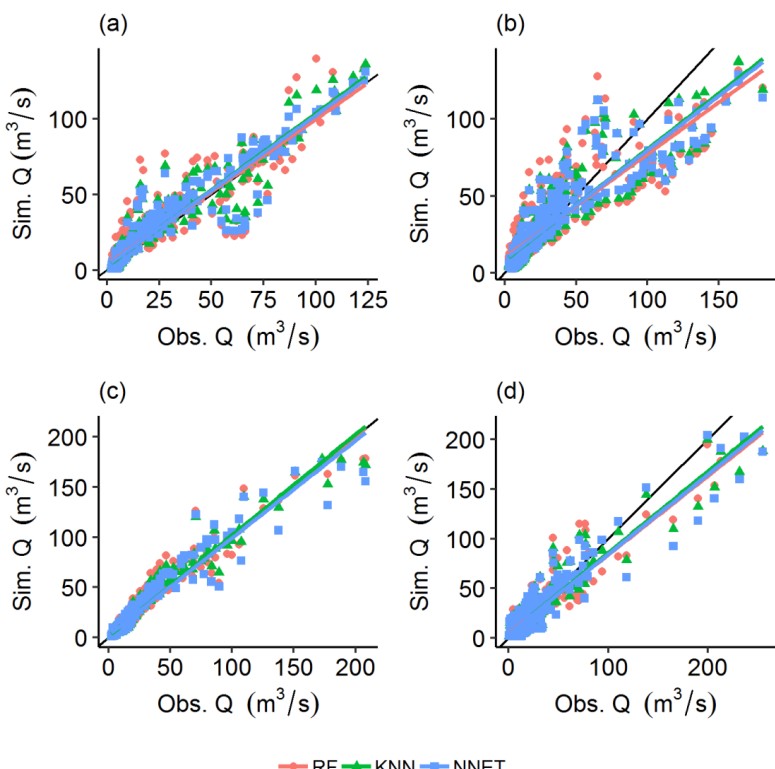

**Figure 3**. Scatterplots of simulated against observed streamflow values for the calibration period, i.e. year 2011 (a) and validation periods, i.e. years 2012 (b), 2013 (c), and 2014 (d). The behavioural models are identified using the coupled MLMs (RF, KNN, and NNET) and GLUE pLoA.

The statistics summarizing the posterior model parameters identified with the help of the three MLMs (RF, KNN, and NNET) and those directly identified from the MC simulation (Calc.) are presented in Table 5. The result shows that the minimum values of $c_1$ and $c_2$ obtained from the three MLMs are similar to the calculated values. Comparable minimum values between the MLMs were also obtained for most of the other parameters, although with slight deviation from the MC estimated values for some of the parameters. For the other statistics, discrepancies were observed both within the MLMs and between the MLMs and MC estimated values. NNET has yielded similar snow coefficient of variation (cv) values as those estimated from the MC simulation for all statistics. However, no consistent result was observed for most of the model parameters. While a certain MLM yields a closer statistics to the calculated values in one parameter, it gets superseded by another MLM in other parameters. Varying degree of distribution characteristics was also observed among the model parameters estimated by a given MLM. For example, $c_3$ and ws have respectively shown highest negative and positive skews of -0.540 and 0.739 as compared to the other parameters when using NNET. In the GLUE methodology, the set of parameters is generally more important than statistical characteristics of the individual parameters since different combinations of the model parameters presented in the table may give similar result. For example, similar streamflow prediction efficiency criteria (NSE, LnNSE, and CR) were obtained during the calibration period of year 2012 when using models identified with the help of RF and NNET (Table 4).





**Table 5.** Statistical summary of posterior distribution for model parameters identified using the coupled MLMs and MC simulation (RF, KNN, and NNET) as well as those directly identified from the MC simulation (Calc.)

| Stat. | MLM | Model parameter | | | | | | | |
|---|---|---|---|---|---|---|---|---|---|
| | | c1 | c2 | c3 | tx | ws | fa | sa | cv |
| Min. | RF | -5.000 | 0.001 | -0.117 | -2.998 | 1.002 | 1.006 | 20.024 | 0.061 |
| | KNN | -5.000 | 0.001 | -0.107 | -2.998 | 1.006 | 1.006 | 20.230 | 0.071 |
| | NNET | -5.000 | 0.001 | -0.115 | -2.998 | 1.006 | 1.006 | 20.024 | 0.060 |
| | Calc. | -5.000 | 0.000 | -0.136 | -2.994 | 1.000 | 1.003 | 20.024 | 0.061 |
| Max. | RF | -3.044 | 0.521 | -0.050 | -0.235 | 3.100 | 14.918 | 39.981 | 0.848 |
| | KNN | -2.511 | 0.509 | -0.050 | 0.592 | 3.777 | 14.772 | 39.981 | 0.849 |
| | NNET | -2.710 | 0.565 | -0.050 | 1.906 | 5.160 | 14.991 | 39.913 | 0.850 |
| | Calc. | -1.766 | 0.845 | -0.050 | 1.980 | 4.205 | 14.991 | 39.990 | 0.850 |
| Mean | RF | -4.182 | 0.172 | -0.078 | -1.858 | 1.963 | 7.440 | 29.942 | 0.437 |
| | KNN | -4.070 | 0.197 | -0.073 | -1.624 | 1.982 | 5.796 | 29.538 | 0.463 |
| | NNET | -4.259 | 0.192 | -0.072 | -1.428 | 2.281 | 8.024 | 30.388 | 0.463 |
| | Calc. | -3.856 | 0.273 | -0.076 | -1.202 | 2.506 | 7.832 | 30.130 | 0.463 |
| Med. | RF | -4.180 | 0.169 | -0.077 | -1.896 | 1.946 | 6.912 | 29.953 | 0.427 |
| | KNN | -4.091 | 0.176 | -0.072 | -1.645 | 1.906 | 5.240 | 29.294 | 0.475 |
| | NNET | -4.341 | 0.168 | -0.069 | -1.594 | 2.132 | 8.046 | 30.803 | 0.470 |
| | Calc. | -3.883 | 0.249 | -0.072 | -1.317 | 2.518 | 7.765 | 30.441 | 0.470 |
| Var. | RF | 0.201 | 0.011 | 0.000 | 0.445 | 0.288 | 16.815 | 35.295 | 0.050 |
| | KNN | 0.373 | 0.017 | 0.000 | 0.640 | 0.377 | 10.216 | 34.891 | 0.041 |
| | NNET | 0.250 | 0.018 | 0.000 | 1.141 | 0.733 | 16.555 | 30.697 | 0.051 |
| | Calc. | 0.415 | 0.035 | 0.000 | 1.263 | 0.672 | 16.250 | 33.799 | 0.051 |
| Skew. | RF | 0.048 | 0.217 | -0.222 | 0.291 | -0.004 | 0.179 | 0.009 | 0.051 |
| | KNN | 0.399 | 0.419 | -0.329 | 0.288 | 0.440 | 0.565 | 0.044 | -0.128 |
| | NNET | 0.568 | 0.541 | -0.540 | 0.685 | 0.739 | -0.033 | -0.077 | -0.055 |
| | Calc. | 0.341 | 0.562 | -0.620 | 0.642 | -0.069 | 0.044 | -0.036 | -0.051 |

### 4.3 Variable importance and interaction

Sensitivity analysis is an important technique to assess the robustness of model based results and it is often performed in
tandem with emulation based studies in order to determine which of the input parameters are more important in influencing
the uncertainty in the model output (Ratto et al., 2012). Figure 4 shows the sensitivity of streamflow predictions to the model
parameters based on the in-built variable importance assessment methods of the three MLMs trained to predict pLoA and
Score. The relative measures of importance are scaled to have a maximum value of 100. The RF and KNN MLMs trained to
predict pLoA yielded similar relative importance of the model parameters. The catchment response parameters of the
hydrological model, *viz.* c1, c2, and c3 have shown higher relative importance as compared to the snow and water balance
parameters. On the other hand, the NNET trained to predict pLoA has yielded higher relative importance for wind scale (ws)
and the rain/snow threshold temperature (tx) as compared to the linear (c2) and quadratic (c3) coefficients of the catchment
response function. The RF and KNN MLMs trained to predict Score have also shown similar result to their equivalent
MLMs trained to predict pLoA with the exception of a swipe in the order of importance between the two least important
parameters, fa and cv, when using RF. The result from the NNET trained to predict Score was less consistent with the result
obtained from its corresponding MLM trained to predict pLoA. The former result was similar to the one obtained from the
KNN trained to predict Score except that c3 was preceded by c1 and ws in the case of NNET. The snow coefficient of
variation (cv) as well as the slow (sa) and fast (fa) albedo decay rates were the least important variables as identified using
the three MLMs when applied to predict pLoA and Score. The relative importance of the model parameters obtained using





the MLMs was generally consistent with the result obtained in previous study focused on parameter uncertainty analysis using the GLUE methodology (Teweldebrhan et al, 2018).

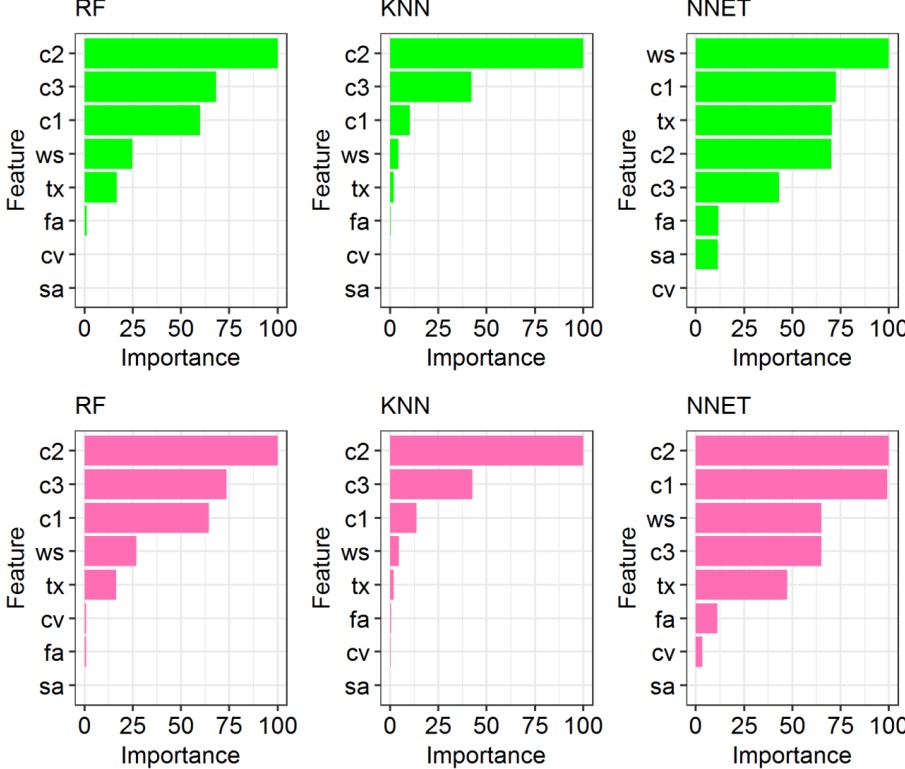

**Figure 4**. Relative importance of the hydrological model parameters based on the three machine learning models, i.e. RF,
5     KNN, and NNET trained for pLoA (upper row) and Score (lower row)

Figure 5 presents a sample correlation matrix of the behavioural model parameters identified using the coupled RF as MLM and the MC simulation. The highest correlation was observed between tx and ws with a Pearson correlation value of 0.57 followed by the correlation between c2 and c3 with a Pearson correlation value of 0.24. A correlation value of 0.22 was also obtained between c1 and ws. The high degree of interaction of ws with tx and c1 reveals that this parameter might have
10    significant effect on model results in combination with the other parameters, although it appears less important when considered alone.





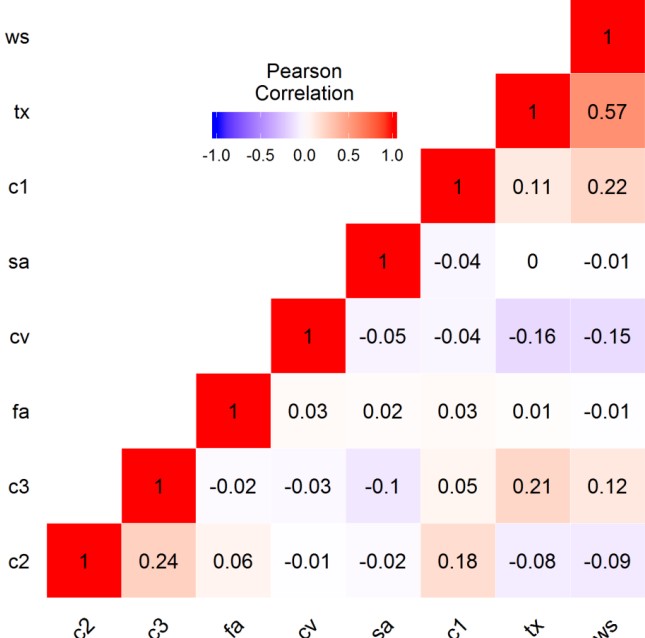

**Figure 5**. Pearson correlation matrix of the behavioural model parameters identified using the coupled RF and the limits of acceptability approach.

## 5  Discussion

The capability of MLMs as emulators of the MC simulation has been demonstrated in this and other similar studies. Machine learning and other data-driven models have been applied as emulators to substitute complex and computationally intensive simulation models. These models have been referred in the literature as surrogate models (e.g. Yu et al., 2015-p20) and metamodels (e.g. El Tabach et al., 2007). Emulators were reported to be particularly useful when a large number of simulations such as the MC simulation are required to be performed, for example, during optimization (Hemker et al., 2008)

and sensitivity analysis (e.g. Reichert et al., 2011). The results from this study revealed that the MLMs trained with limited sample size of artificially generated data from the simulation model were computationally efficient and providing reliable approximation of the underlying hydrological system. Similar advantages of MLM based emulators were also reported in previous studies (e.g. Kingston et al., 2008; Razavi et al., 2012).

The MLMs applied in this study and in other areas of application have both advantages and limitations. MLMs are able

to learn complex nonlinear system from a set of observations and usually yielding a high degree of accuracy as they are not affected by the level of understanding of the underlying processes in the system (Kingston et al., 2008). Furthermore, MLMs with the virtue of their generalization capability are relatively quick to run as simulations over an extended period of time are not required. However, since MLMs do not have any understanding of the modelled physical processes, they operate as black-box models with an accompanying dilemma on whether they would behave as intended under changing future

conditions (Olden and Jackson, 2002). Generally, MLMs have limited application in conditions that significantly deviate from historical norms. In this study, adequate size of training samples was used in order to represent different parts of the





parameter dimensions. Furthermore, in many MLMs the notion of degrees of freedom is usually ignored when computing performance metrics during model training (Kuhn, 2008). Since these metric do not penalize model complexity (e.g. as in the case of adjusted $R^2$), they tend to favour more complex fits over simpler models. In some MLMs a regularization approach is employed to adjust the cost function in such a way that the model learns slowly and thereby minimize overfitting (Nielsen, 2018). In this study, for example, the L2 regularization was used with the NNET model.

In studies involving use of coupled ML and MC simulation, the uncertainty in parameter identification may stem from various sources. For example, the relative mismatch between the observed and simulated streamflow for the validation period in years 2012 and 2014 as compared to the good fit in year 2013 (Fig. 3) can be attributed to the differences in hydrological conditions between the calibration and validation periods. Figure 6 shows the observed streamflow values of the four hydrological years at different percentiles. As can be noticed from this figure, the observed streamflow values for the validation period in year 2012 exceed those for the calibration period (Year 2011) at all percentile values. On the other hand, streamflow recorded in year 2013 have shown closer values to those from year 2011 at most of the percentiles. The result from this analysis reveals that the identified model parameters yielded lower performance when applied to a hydrological condition that significantly deviated from the observations used for the identification of these parameters. This can be due to the prevalence of different dominant processes in different hydrological conditions.

The highest average NSE and LnNSE for the validation periods were obtained when using models identified in year 2012 and year 2014, respectively (Table 4). The Nash-efficiency computed using the row streamflow data (NSE) gives more emphasis to high-flow than low-flow values, while the one computed using the log-transformed data (LnNSE) gives more emphasis to low-flow conditions. Thus, the models identified under the predominantly low-flow condition, i.e. year 2014 were good on predicting low-flows while those identified under high-flow condition, i.e. year 2012 were good in the prediction of high-flows when applied during the validation period. Generally, these phenomena are consistent with concerns raised in previous studies focused on the challenges of the model development philosophy based on a universal fixed model structure that is transposable in both space and time (e.g. Clark et al., 2011; Kavetski and Fenicia, 2011). The results from this and other similar studies (e.g. Fenicia et al., 2011) suggest the need for additional components to emphasize on dominant processes, although fixed model structures might be attractive due to their relatively parsimonious structure.

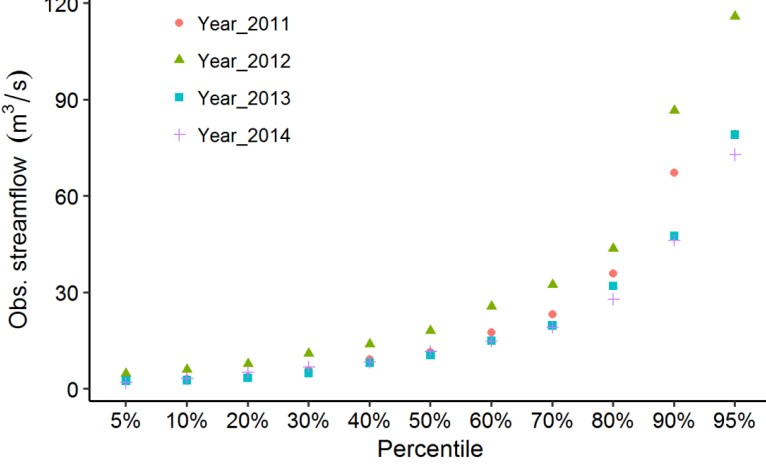

**Figure 6**. Comparison of the percentile observed streamflow values for the calibration period (Year_2011) and validation periods (Year_2012, Year_2013, and Year_2014)





Although KNN was not a favourite emulator in previous hydrological studies, it has yielded a comparable result to the other MLMs used in this study. For example, the performance of KNN was superior to RF and NNET based on the average NSE obtained for the calibration period. However, the result from KNN was characterized by higher inter-annual variability as compared to RF and NNET. Inconsistent relative performances between KNN and NNET were also reported in previous

studies focused on flow forecasting using MLMs. For example, Wu and Chau (2010) obtained a better monthly streamflow forecast using KNN as compared to NNET, although Mekanik et al. (2013) observed better performance of NNET as compared to KNN. A similar inconsistent result was also observed in another study focused on monthly streamflow forecasting with a higher cumulative ranking of NNET as compared to KNN under nonlinear conditions (Modaresi et al., 2018). However, the later was better in reproducing the observations under linear condition; and they concluded that the

variability in relative performance of the MLMs may be attributed to the differences between study sites, data sets, and structure of the MLMs as well as whether the relationship between the predictor and predicted variables is linear or nonlinear. The main challenges with KNN appear when data are sparse, although this problem can be partly overcome by choosing the number of neighbours adapted to the concentration of the data (Burba et al., 2009).

In this study, different trials were conducted in order to assess effects of the model structure and hyper-parameter values

and thereby to get the optimal MLMs (result not shown). For example, the NNET model with multiple hidden layers resulted to lower performance than the one with single hidden layer. This result is consistent with the general notion, that for many applications a single hidden layer is adequate to model any nonlinear continuous function (e.g. Hsieh,2009; Snauffer, et al., 2018). Similarly, use of a linear activation function has yielded NNET models with better accuracy as compared to the commonly used sigmoidal function.

In previous emulator based uncertainty analysis studies, the residual based GLUE methodology was coupled with the MLMs (e.g. Yu et al., 2015). Here, we used the limits of acceptability concept in order to overcome some of the limitations associated with the residual based approach. The original formulation of the GLUE LoA is, however, too strict for use in identification of behavioural models and it may result to rejection of useful models and thereby making type II error. In order to minimize such errors, one of the commonly used approaches was to relax the limits (e.g. Blazkova and Beven, 2009).

However, in previous study it was observed that relaxing the limits was not a feasible option in simulations that involve time series data with dynamic observational error characteristics as in the case of continuous rainfall-runoff modelling. Accordingly, in an attempt to balancing between type I and type II errors, the time-relaxed limits of acceptability approach was introduced (Teweldebrhan et al., 2018). This approach was employed in this study and it relaxes the strict criterion of the original formulation that demands all model predictions to fall within their respective observation error bounds. When

using this approach, the minimum threshold for the percentage of time steps where model predictions are expected to fall within the limits is defined as a function of the level of modelling uncertainty.

A combined likelihood measure based on the persistency of model realizations in reproducing the observations within the observational error bounds (pLoA) and a normalized absolute bias (Score) was used in previous study focused on snow data assimilation (Teweldebrhan et al., 2019). The Score values were rescaled with due consideration to pLoA, whereby the two

efficiency measures were given equal importance in estimating the final weight of each model. In this study, the acceptable models were first identified based on pLoA only and the Score was used to weigh the relative importance of the acceptable models in predicting the quantile streamflow values. Another trial that involved selection of the top 100 best performing models using a combined likelihood with equal weights given to pLoA and Score yielded relatively low validation result as compared to using pLoA alone for the identification of behavioural models (result not shown). This can be attributed to the

difference in nature of these likelihood measures. pLoA considers only the percentage of time steps where the model





predictions have fallen within the observation error bounds. This renders pLoA to be less sensitive to the variability in relative performances of the model between time steps. On the other hand, Score can be highly affected by predictions of few time steps that are very close or too far from the observed value, albeit within the limits.

## 6   Conclusions

Three machine learning models (MLMs), i.e. Random forest (RF), K-Nearest Neighbours (KNN), and an Artificial Neural-Network (NNET) were constructed to emulate the time consuming MC simulation and thereby overcome its computational burden when identifying behavioural parameter sets for a distributed hydrological model. Two sets of MLMs were trained using the randomly generated uncertain model parameter values as covariates, and two efficiency criteria defined within the realm of the limits of acceptability concept as target variables. One of the efficiency criteria used in this study was a measure of model persistency in reproducing the observations within the observation error bounds (pLoA), while the other one was based on a normalized absolute bias (Score).

The coupled MLMs and time-relaxed limits of acceptability approach employed in this study were able to effectively identify behavioural parameter sets for the hydrological model. The MLMs were able to adequately reproduce the response surfaces for the test and validation samples, although the evaluation metrics have shown variability both between the MLMs and the analysis years. RF and NNET yielded comparable results (especially for pLoA), while KNN has shown relatively lower result. Capability of the MLMs as emulators of the MC simulation was further evaluated through comparison of streamflow predictions using the identified behavioural model realizations against the observed streamflow values. The cross-validation result shows that the high-flow conditions as measured by average NSE were slightly better estimated both under the calibration and validation periods when KNN was used as emulator as compared to RF and NNET, while NNET yielded a slightly better prediction under low-flow conditions (LnNSE). Although the behavioural models identified based on KNN have shown a relatively higher inter-annual variability, they have yielded comparable performance to RF and NNET in terms of the efficiency measures. Future studies may assess the possibility of using the three MLMs as ensemble emulators to get an improvement in the identification of behavioural parameter sets while significantly minimizing the computational burden of the MC simulation.

The sensitivity analysis conducted using the in-built algorithms of the three MLMs have yielded comparable order of precedence in relative variable importance when trained using pLoA and Score as target variables. The result was generally consistent with the one obtained from previous study conducted using the residual-based GLUE methodology. The catchment response parameters of the hydrological model, i.e. $c_1$, $c_2$, and $c_3$ have shown higher relative importance as compared to the snow and water balance parameters. Thus, the efficiency of MLM based emulators in doing sensitivity analysis for computationally expensive models was also further proven in this study.

*Data availability*. The underlying hydrologic observations for this analysis were provided by Statkraft AS and are proprietary within their hydrologic forecasting system. However, the data may be made available upon request.

*Competing interests*. The authors have no conflict of interest.

*Acknowledgements*. This work was conducted within the Norwegian Research Council's - Enhancing Snow Competency of Models and Operators (ESCYMO) project (NFR no. 244024) and in cooperation with the strategic research initiative





LATICE (Faculty of Mathematics and Natural Sciences, University of Oslo https://mn.uio.no/latice). We thank Statkraft AS for providing us the hydro-meteorological data.

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
