# Peer review of "Coupled machine learning and the limits of acceptability approach applied in parameter identification for a distributed hydrological model"

_Hydrology and Earth System Sciences, 2019_

## Referee Comment (RC1) · Anonymous Referee #1 · 14 Nov 2019

**Summary**

In this manuscript, the authors employ a specific set of machine-learning based emulators (random forests, kNN and ANN) for parameter and uncertainty estimation while using two different parameter identification metrics: (a) the absolute bias score of model predictions and (b) GLUE with time relaxed limits of acceptability (pLoA). The results from the work are mentioned in the abstract: "The three MLMs (machine-learning models) were able to adequately mimic the response surfaces directly estimated from MC simulations; and the models identified using the coupled ML emulators and the limits of acceptability approach have performed very well in reproducing the median streamflow prediction both during the calibration and validation periods."

**General impression of the reviewer**

While both (a) unbiased/improved parameter identification and (b) fast emulation of expensive hydrologic simulators are important research topics, I have major reservations with regard to the analysis and conclusions of this manuscript. The authors have already published a paper on the performance of ploA (Teweldebrhan et al., 2018) and there are several detailed papers/studies out there on the performance of various emulation techniques as cited by the authors (page 2, last paragraph, page 3 first paragraph). Combining these two separate research questions makes it difficult to judge whether the presented analysis is targeting the performance of ploA as a parameter/uncertainty estimation metric or of these three machine-learning techniques as efficient emulators. Such a joint treatment of these two separate questions doesn't necessitate any adequately new insight into either the emulator efficiency or parameter estimation. To what does one ascribe this conclusion - ploA or emulation?: "ML emulators and the limits of acceptability approach have performed very well in reproducing the median streamflow prediction both during the calibration and validation periods."

Additionally, while this manuscript can be a useful addendum of information for those interested in parameter estimation using the GLUE pLoA approach, it doesn't seem to qualify as a stand-alone novel application. The principal criteria for the acceptance of a paper in HESS are: **Scientific significance, Scientific quality, and Presentation quality**: I am afraid that as far as the first two criteria are concerned, the manuscript does not do better than a "fair" classification.

Therefore, I am inclined to reject this paper, and suggest a resubmission in a more suitable journal. However, in case other reviewers differ in their judgement, I request the authors to make these substantial revisions, in both the analysis and the writing structure, based on the following comments before it can be considered for publication in HESS.

**Specific comments**

1. A good emulator ( in this case a mapping between $\mathbb{R}^8 \to \mathbb{R}$) may not help to improve the streamflow predictions if the identification metric or the hydrologic models are bad. So the performance of emulation is a somewhat independent question from that of the performance of an identification metric. From the manuscript, the conclusions suggest that both emulation and pLoA together happen to work well. But even that is doubtful as the paper does not comment on many aspects of emulation. (a) How do these techniques perform when the models are run fewer number of times, say only 400 times instead of 4000? (b) How do these techniques perform with a parameter space of higher dimensionality (n) such that $\mathbb{R}^n \to \mathbb{R}$? (c) Also, what is the added utility of the 95000 simulations in comparison to the already 4000 runs? Any recommendations/comments on the number of samples required for convergence? (d) How does the emulator perform in extrapolation phase (the 80% calibration, 20% validation separation will not be adequate to show how the emulator may diverge when one uses parameter values away from the training data set. This implications will be more severe when the emulators are used in Bayesian inference and the prior distribution of parameters is not hard-bounded). (e) And perhaps analysing or commenting on the time efficiency of emulators.

2. What new insights do we get from the application of emulation tools to this pLoA metric, apart from the fact that it is a possibility to emulate? "the three MLMs were able to adequately mimic the response surfaces directly estimated from MC simulations" This needs to be made clear (preferably using numbers) in the abstract, discussion and conclusions.

3. What is the interpretation of the output generated from behavioral parameters? Do we expect the observations to lie within these bands with a certain frequency? (please refer to Stedinger et al. 2008, for more insights on this debate) If yes, then the reader would like to see reliability (q-q) plots to gauge the performance.

4. How much of the statements made about the efficiency of the emulator are dependent on the choice of the specifications of those machine learning techniques? A paragraph on the meta parameters of this study will be appreciated.

5. Some hydrographs will be a useful addition to the existing plots.

6. Please explain why an assumption of 25% for observation error and what will be the effect of choosing a different value on the performance of either GLUE pLoA and the emulation.

---

## Referee Comment (RC2) · Anonymous Referee #2 · 13 Jan 2020

REVIEW of the paper

**Coupled machine learning and the limits of acceptability approach applied in parameter identification for a distributed hydrological model**

Authors: Aynom T. Teweldebrhan, John F. Burkhart, Thomas V. Schuler, Morten Hjorth-Jensen
Manuscript Number: hess-2019-464

Submitted: **HESSD**

This paper presents machine learning methods (MLMs) to emulate MC simulations to identifying behaviour parameter sets of hydrological model. Three MLMs were trained on limited number of MC samples to predict some sort of error or loss function of the MC simulations. Trained models were then used to predict loss function for a large number of samples from which the behavioural parameter sets were identified. While the results look reasonable, there are two main fundamental issues in this manuscript. Authors claimed that the proposed method overcomes computational burden of MC simulations and subjectivity in choosing the likelihood and the threshold value in GLUE. Manuscript fails to provide sufficient evidence to support both claims (see comments below).

I am struggling to find main motivation of this work. It is mentioned that emulators are used to minimize the computational burden of the MC simulation. But this is not completely true. Emulators are used only to predict some sort of likelihood values of the simulation to know whether it should be rejected or not in GLUE framework. Then hydrological models are run with behavioural parameter sets to quantify predictive uncertainty. In other words, MC simulations are still used. Indeed, the proposed method does not save computational time when it is required e.g., in real time forecast. For example flood emergency managers want to know the probability of exceeding major flood level at tomorrow noon. There are other ways to emulate MC simulations which are saving computational time in real time application (e.g., Shrestha et al., 2009; Shrestha et al., 2014).

Another issue in this manuscript is that proposed GLUE pLoA is not convincing. Authors mentioned that the original GLUE has issue in subjectively choosing a likelihood and threshold value for identification of behavioural and non-behavioural parameter sets. They proposed GLUE pLoA to overcome these limitations, however it introduces two additional settings to choose: error bounds and percentage of the model predictions that fall within the error bounds to identify whether given simulation is behavioural and non-behavioural. So proposed method is also subjective, indeed more complex than the original GLUE and requires iterations to choose percentage of the model predictions that fall within the error bounds that satisfy the acceptable CR value.

Verification scores used in this manuscript do not directly test accuracy of emulators to identify behavioural or non-behavioural parameters sets. In this manuscript, RMSE and

related measures were used as performance measures of the emulators. However, the problem should be formulated as classification rather than regression if the objective of emulators is to identify whether given simulation is behavioural or non-behavioural. Classification problem is very straightforward:

- Classify each MC simulation to behavioral or non-behavioral model using GLUE pLoA
- Train and test emulators to classify whether given MC simulation is behavioral or non-behavioral model
- Verify the emulators to test accuracy of the classification using a 2 by 2 contingency table similar to used in weather forecast. In this table "hit" represents number of the cases when the MLM correctly identifies or classifies the behavioral parameter sets (i.e. classification from MLM is behavioral for behavioral parameter sets). From this table it is possible to compute various scores including hit rates (Hits/(Hits+Misses) etc.

| MLM Emulators | Parameter Sets | |
|---|---|---|
| | Behavioural | Non-Behavioural |
| Behavioural | Hits | False alarms |
| Non-Behavioural | Missed | Correct negatives |

**Minor comments**

P3, L32: define Score.

P4, L14: What is the basis for 25% as mean observational uncertainty? It is not clear how streamflow limits are computed using this observation uncertainty. Since hydrological model errors are heteroscedastic, applying same value of 25% of the mean observation as error bounds for all time steps would be problematic.

P4, L27: Define acceptable pLoA. Is it CR from the original GLUE? I wonder what GLUE CR value is. I think this is another subjectivity in this method. Importantly the proposed method relies on original GLUE method to identify acceptable CR. In other words, the proposed GLUE pLoA is not completely independent method, it relies on residual GLUE method to compute its hyper parameters such as acceptable CR.

P4, Step 3: "… specified percentage of the total observations." Here is one of subjectivity to identify whether the model simulation is behavioral or non-behavioral. What value is used?

P5, L1: Equation 2 should be defined before steps.

P5, L9: Since all terms of Equation 3 are not defined (e.g. $l, u$) and assuming $L_e$ in this equation is same as $L_e$ defined in equation 2, I am not sure if the equation is correct. It is not clear whether $e$ is absolute. In either case, for example first expression $\mu_Q = 0, e \leq L_e$

might not be correct. It is better to illustrate Equation (3) with a figure similar to the following

$$\mu_Q = \begin{cases} 0, Q_{sim} \leq L_e \\ \dfrac{Q_{sim} - L_e}{Q_{obs} - L_e}, L_e < Q_{sim} \leq Q_{obs} \\ \dfrac{U_e - Q_{sim}}{U_e - Q_{obs}}, Q_{obs} < Q_{sim} \leq U_e \\ 0, Q_{sim} \geq U_e \end{cases}$$

P6, Line 31: 5000 samples may not truly represent the parameter uncertainty. I suggest to use convergence analysis to know the number of samples.

P11, L5, what is the validation data set? Is it S3?

P13, Table 4: Another widely used cross-validation method is leave out cross-validation. For example, for leave-one-year-out cross-validation, generate simulations in 2011 using model calibrated (e.g., behavioral parameter sets identified) in all data except year 2011, generate simulations in 2012 using model calibrated in all data except year 2012 and so on. Then all simulation data from year 2011, 2012, 2013, and 2014 can be collated to verify the results. This cross validation procedure is expected to produce results that are comparable to those obtainable under operational conditions as the number of data used to fit the model will be similar to that available for operational applications.

P15, Table 5: I strongly suggest replacing Table 5 with distribution plots which is more readable.

P15, L3: Section 4.3 is not relevant to this study, so can be deleted.

P18, l17, row?

**References**

Shrestha, D.L., Kayastha, N., Solomatine, D., 2009. A novel approach to parameter uncertainty analysis of hydrological models using neural networks. Hydrol. Earth Syst. Sci., 13: 1235-1248.

Shrestha, D.L., Kayastha, N., Solomatine, D., Price, R., 2014. Encapsulation of parametric uncertainty statistics by various predictive machine learning models: MLUE method. Journal of Hydroinformatics, 16(1): 95-113.

---

## Author Comment (AC1) · 10 Feb 2020

Reply to the general impression of the reviewer

Dear reviewer, as you have pointed out under the specific comments (1), the identification of behavioural models through coupling of emulators is affected by multiple factors. It depends on nature of the likelihood measure and its predictability as independent variable (for example in this study, between pLoA and Score). It also depends on the type of fitting model (emulator) used to estimate value of the likelihood measure (in this case the machine learning models).

Although residual-based likelihood measures were used in previous similar studies, as of our best knowledge none of **the emulator based studies** have used pLoA or Score as a response surface, and the limits of acceptability approach in general. And it is for this reason that the first objective of this study was focused on assessing the possibility of using pLoA for the identification of behavioural models using the **coupled** MLMs and the limits of acceptability approach. Further, since the three machine learning models are applied to predict the same response variables followed by the identification of behavioural models using the limits of acceptability approach, the relative performance of RF and KNN (that were not applied in previous studies) can be easily evaluated against the standard ML model, i.e. NNET. And this forms the basis for the second objective of this study, for which the authors believe gives a new insight into the possibility of using RF and KNN as emulators of the MC simulation for application in parameter identification.

To what does one ascribe this conclusion - ploA or emulation?: "ML emulators and the limits of acceptability approach have performed very well in reproducing the median streamflow prediction both during the calibration and validation periods."

The median streamflow prediction is the result from the **coupled** effect of both the likelihood measure (pLoA) and the specific emulator used to predict the likelihood values.

1. A good emulator (in this case a mapping between $\mathbb{R}^n \to \mathbb{R}$?) may not help to improve the streamflow predictions if the identification metric or the hydrologic models are bad. So the performance of emulation is a somewhat independent question from that of the performance of an identification metric.

This comment is consistent with the response provided above for "the general impression of the reviewer".

From the manuscript, the conclusions suggest that both emulation and pLoA together happen to work well. But even that is doubtful as the paper does not comment on many aspects of emulation.

(a) How do these techniques perform when the models are run fewer number of times, say only 400 times instead of 4000?

Thank you, we will accommodate this comment in the revised version. A preliminary analysis using 400 samples shows that some of the coupled emulators fail to produce any behavioural model in certain years.

(b) How do these techniques perform with a parameter space of higher dimensionality (n) such that $\mathbb{R}^n \to \mathbb{R}$?)?

Sensitivity of the emulation-based parameter identification to parameter space dimension was not conducted since running the hydrological model used in this study under a distributed setting requires a long time. The model is structured in such a way that, at each time step, the main processes of the model run on each of the grid-cells. This challenge becomes more pronounced when we consider the need for high number of model runs in order to overcome the non-identifiability problem for high parameter space dimensions. Thus, the assessment for effect of parameter space on emulation-based parameter identification might be the subject of our future studies.

(c) Also, what is the added utility of the 95000 simulations in comparison to the already 4000 runs? Any recommendations/comments on the number of samples required for convergence?

Like most studies based on the GLUE methodology, the main focus of this study was also to get as much behavioural models as possible so as to encapsulate future uncertain conditions. However, only little to no improvement was obtained in most cases when assessed using the available evaluation dataset and the streamflow evaluation metrics used in this study.

(d) How does the emulator perform in extrapolation phase (the 80% calibration, 20% validation separation will not be adequate to show how the emulator may diverge when one uses parameter values away from the training data set. This implication will be more severe when the emulators are used in Bayesian inference and the prior distribution of parameters is not hard-bounded).

As presented in the manuscript (Validation columns in Table 3), capability of the emulators to reproduce the response surface generated directly from the Monte Carlo simulations was further assessed using the 95, 000 samples (S3) in addition to the 20% (test) samples.

(e) And perhaps analysing or commenting on the time efficiency of emulators.

We will accommodate this comment in the revised version of the manuscript. The emulators normally take few seconds to generate the response surfaces for the 95000 samples. And when it comes to the Monte Carlo simulation, it was assumed that each of the iterations requires same amount of time. Accordingly, the amount of time required would be proportional to the number of iterations.

2. What new insights do we get from the application of emulation tools to this pLoA metric, apart from the fact that it is a possibility to emulate?

Since the predictability of independent variables varies from one to another, application of emulation methods to predict pLoA gives us a further insight on the potential and scope of the emulators to predict different response surfaces in addition to the residual-based likelihood measures that were applied in previously studies.

"the three MLMs were able to adequately mimic the response surfaces directly estimated from MC simulations". This needs to be made clear (preferably using numbers) in the abstract, discussion and conclusions.

The detailed result supporting this conclusion is presented in Table 3 and explained in section 4.1 of the manuscript. As suggested, we will also provide some metric values in the abstract, discussion, and conclusions in the revised version of the manuscript.

3. What is the interpretation of the output generated from behavioral parameters? Do we expect the observations to lie within these bands with a certain frequency? (please refer to Stedinger et al. 2008, for more insights on this debate) If yes, then the reader would like to see reliability (q-q) plots to gauge the performance.

Thank you for your suggestion to the reading material. It provides further insight on uncertainty analysis in hydrological modelling. This theme has been the subject of debate in many hydrology literatures. In order to avoid any confusion with the confidence level expected from the formal Bayesian approach, we will include the following text in the revised version:

When using the GLUE methodology, the observations are not expected to lie within the prediction bands at a percentage that equals the given certainty level. However, the modeller can adopt the certainty level specified for producing the prediction limits as a kind of standard for assessing the efficiency of the prediction limits in enveloping the observations (Beven, 2006).

4. How much of the statements made about the efficiency of the emulator are dependent on the choice of the specifications of those machine learning techniques? A paragraph on the meta parameters of this study will be appreciated.

As suggested, we will include a paragraph on hyper-parameters of the machine learning models in the revised version of the manuscript.

5. Some hydrographs will be a useful addition to the existing plots.

As suggested, hydrograph plots will be included in the revised version of the manuscript.

6. Please explain why an assumption of 25% for observation error and what will be the effect of choosing a different value on the performance of either GLUE pLoA and the emulation.

In the GLUE LoA methodology, the limits are set with due consideration to the observation and input errors. Since observational error values were not available for the study area, this value was set based on literature value and observations from a neighbouring catchment plus assumed allowance for input errors. In our previous study, a preliminary assessment on effect of relaxing the limits further, i.e. over 25% while keeping the threshold pLoA at 100% have yielded to the inclusion of non-behavioural models, leading to very low performance during the validation period.

**References:**

Beven, K.: A manifesto for the equifinality thesis. Journal of Hydrology, 320, 2006.

Choi, H. T. and Beven, K.: Multi-period and multi-criteria model conditioning to reduce prediction uncertainty in an application of TOPMODEL within the GLUE framework, Journal of Hydrology, 332, 2007.

Kirchner, J. W.: Catchments as simple dynamical systems: Catchment characterization, rainfall-runoff modeling, and doing hydrology backward, Water resources research, 45, 2009.

Ratto, M., Castelletti, A., and Pagano, A.: Emulation techniques for the reduction and sensitivity analysis of complex environmental models, Environmental Modelling & Software, 34, 2012.

Stedinger, J. R., Vogel, R. M., Lee, S. U., and Batchelder, R.: Appraisal of the generalized likelihood uncertainty estimation (GLUE) method, Water resources research, 44, 2008.

---

## Author Comment (AC2) · 10 Feb 2020

This paper presents machine learning methods (MLMs) to emulate MC simulations to identifying behaviour parameter sets of hydrological model. Three MLMs were trained on limited number of MC samples to predict some sort of error or loss function of the MC simulations. Trained models were then used to predict loss function for a large number of samples from which the behavioural parameter sets were identified. While the results look reasonable, there are two main fundamental issues in this manuscript. Authors claimed that the proposed method overcomes computational burden of MC simulations and subjectivity in choosing the likelihood and the threshold value in GLUE. Manuscript fails to provide sufficient evidence to support both claims (see comments below).

I am struggling to find main motivation of this work. It is mentioned that emulators are used to minimize the computational burden of the MC simulation. But this is not completely true. Emulators are used only to predict some sort of likelihood values of the simulation to know whether it should be rejected or not in GLUE framework. Then hydrological models are run with behavioural parameter sets to quantify predictive uncertainty. In other words, MC simulations are still used. Indeed, the proposed method does not save computational time when it is required e.g., in real time forecast. For example flood emergency managers want to know the probability of exceeding major flood level at tomorrow noon. There are other ways to emulate MC simulations which are saving computational time in real time application (e.g., Shrestha et al., 2009; Shrestha et al., 2014).

Another issue in this manuscript is that proposed GLUE pLoA is not convincing. Authors mentioned that the original GLUE has issue in subjectively choosing a likelihood and threshold value for identification of behavioural and non-behavioural parameter sets. They proposed GLUE pLoA to overcome these limitations, however it introduces two additional settings to choose: error bounds and percentage of the model predictions that fall within the error bounds to identify whether given simulation is behavioural and non-behavioural. So proposed method is also subjective, indeed more complex than the original GLUE and requires iterations to choose percentage of the model predictions that fall within the error bounds that satisfy the acceptable CR value.

Dear reviewer, as mentioned in the manuscript, only 5000 MC simulations are run instead of the 95000 from which the behavioural models are identified. The emulators normally take few seconds to predict the response surfaces for the 95000 samples. And this justifies how much the computational cost has reduced as a result of using the MLMs to predict the response surface for the 95000 samples instead of using MC simulations.

As mentioned in the manuscript (Page 2, line 13; Page 4, line 22), GLUE pLoA is a time-relaxed variant of GLUE LoA which was introduced in our previous study (Teweldebrhan et al., 2018). Thus, the main goal of this study is to minimize the computational cost when using GLUE pLoA rather than proposing the methodology or comparing against other variants of the GLUE methodology. But we would like to reiterate that it was proposed as part of the endeavour to minimize the rejection of useful models when using the original GLUE LoA formulation rather than to dealing with the subjectivity. Useful models were effectively identified using GLUE pLoA, while all of the 100000 simulations were rejected as non-behavioural models when using the original GLUE LoA formulation (Teweldebrhan et al., 2018).

Verification scores used in this manuscript do not directly test accuracy of emulators to identify behavioural or non-behavioural parameters sets. In this manuscript, RMSE and related measures were used as performance measures of the emulators. However, the problem should be formulated as classification rather than regression if the objective of emulators is to identify whether given simulation is behavioural or non-behavioural.

As indicated in the manuscript (e.g. page 3, lines 33-35), the emulators were used to predict the response surfaces for new parameter sets. The identification of behavioural models is, however, a result from the **coupled effect** of the emulators in reproducing the response surfaces and the GLUE pLoA in identifying the behavioural parameter sets. Thus, first capability of the emulators to reproduce the response surface was evaluated through comparison of the predicted against MC simulation based values. Then, performance of the behavioural models was evaluated through comparison of their streamflow simulation result against observed values.

We appreciate for the alternative insight you provided us to dealing with the problem. However, in GLUE pLoA, the models are evaluated as ensemble, based on their capability to produce a CR value close to the predefined value, rather than as individual models. For this reason estimating the response surface using a regression method was found to be more relevant than generating binary values (behavioural/non-behavioural) using classification algorithms.

P3, L32: define Score.

This term was defined earlier in Page 3, line 20

P4, L14: What is the basis for 25% as mean observational uncertainty? It is not clear how streamflow limits are computed using this observation uncertainty. Since hydrological model errors are heteroscedastic, applying same value of 25% of the mean observation as error bounds for all time steps would be problematic.

Since no stage-discharge relationship exists for estimating the streamflow uncertainty using the usual practice, i.e. by fitting different rating curves, an assumed value of 25% was adopted based on certain literature values and observational errors analysed for a neighbouring catchment. This value also takes into account incommensurability and uncertainty in the input dataset. The streamflow observational error bounds (limits) of each observation are estimated as ±25% of the corresponding observation, instead of the mean observation. Yet, as the reviewer mentioned since model errors are heteroscedastic mainly in response to the variability in input dataset errors, it would be too strict to expect a given model to satisfy the limits of acceptability criteria in 100% of the observations. And it is this phenomenon that has called the need to introduce the time relaxed variant of the original GLUE LoA formulation.

P4, L27: Define acceptable pLoA. Is it CR from the original GLUE? I wonder what GLUE CR value is. I think this is another subjectivity in this method. Importantly the proposed

method relies on original GLUE method to identify acceptable CR. In other words, the proposed GLUE pLoA is not completely independent method, it relies on residual GLUE method to compute its hyper parameters such as acceptable CR.

As indicated in P4, L32, the acceptable pLoA is the one that yields a calculated CR value close to the predefined acceptable CR value.

As mentioned in line P4, L17, The CR value is expressed as the number of observations falling within their respective prediction bounds to the total number of observations (Eq. 1). In this study, the CR value obtained using the residual based GLUE methodology was used for the ease of comparing the result obtained from both methodologies. However, the modeller may also set the acceptable CR value based on previous experience, although this involves some degree of subjectivity.

P4, Step 3: "… specified percentage of the total observations." Here is one of subjectivity to identify whether the model simulation is behavioral or non-behavioral. What value is used?

The iteration to get an acceptable pLoA value starts from 100% and decreases further, i.e. relaxed until the desired level of CR is achieved. The reason for relaxing this criterion is provided under the response to the P4, L14 comment. We would, however, like to iterate that relaxation in the GLUE LoA approach in order to overcome the rejection of useful models is not a new phenomenon. The difference with the previous approaches lies on use of the time relaxed approach than, for example, extending the limits (e.g. Choi and Beven, 2007).

P5, L1: Equation 2 should be defined before steps.

As suggested, we will do that in the revised version of the manuscript.

P5, L9: Since all terms of Equation 3 are not defined (e.g. $l,u$ ) and assuming $L_e$ in this equation is same as $L_e$ defined in equation 2, I am not sure if the equation is correct. It is not clear whether $e$ is absolute. In either case, for example first expression $\mu_Q = 0, e \leq L_e$ might not be correct. It is better to illustrate Equation (3) with a figure.

Thank you, the notations $l$ and $u$ respectively correspond to $L_e$ and $L_u$. Thus, we will replace them with the latter notations in order to be consistent with the notations in Equation 2. We will also provide an illustrative figure accompanying Equation 3 similar to the suggested one.

The notation $e$ is not absolute and thus the expression $\mu_Q = 0, e \leq L_e$ is correct, since a model producing a negative error value of less than the lower observational error bound (which is also a negative value) has 0 degree of membership.

P6, Line 31: 5000 samples may not truly represent the parameter uncertainty. I suggest to use convergence analysis to know the number of samples.

The 5000 samples were used for training and testing of the machine learning emulators. While the behavioural parameter sets that actually are less than 5000 were identified from the 95000 samples (Section 2.3). The reason for the low number of behavioural samples is partly attributed to the use of uniform parameter distribution and the simple Monte Carlo method for parameter sampling. However, analyses conducted using 50000 and 100000 samples in our previous study have yielded similar parameter and streamflow uncertainty results. We will include a text about this in the discussion session of the revised manuscript.

P11, L5, what is the validation data set? Is it S3?

Here the validation dataset refers to S3 and the corresponding response surface values estimated using the MC simulations. We will clarify this in the revised manuscript.

P13, Table 4: Another widely used cross-validation method is leave out cross-validation. For example, for leave-one-year-out cross-validation, generate simulations in 2011 using model calibrated (e.g., behavioral parameter sets identified) in all data except year 2011, generate simulations in 2012 using model calibrated in all data except year 2012 and so on. Then all simulation data from year 2011, 2012, 2013, and 2014 can be collated to verify the results. This cross validation procedure is expected to produce results that are comparable to those obtainable under operational conditions as the number of data used to fit the model will be similar to that available for operational applications.

Thank you for the suggestion to the alternative cross-validation method. In this study we have preferred to test the model using the worst case scenario, i.e. if we have only one hydrological year for model calibration. Further, as presented in the discussion section, this method allows us to examine the performance of models identified in a given hydrological year when applied under a highly different hydrological condition. A similar approach was used in previous hydrological studies; and it was considered as a more rigorous validation method than the commonly used split-sample methods (e.g. Kirchner, 2009).

P15, Table 5: I strongly suggest replacing Table 5 with distribution plots which is more readable.

Thank you for the suggestion. We will replace this table with distribution plots displaying the distribution of each parameter under the different emulators.

P15,L3: Section 4.3 is not relevant to this study, so can be deleted.

As discussed in previous studies (e.g. Ratto et al., 2012), sensitivity analysis is often performed in tandem with uncertainty analysis in order to determine which of the input parameters are more important in influencing the uncertainty in the model output. Conducting sensitivity analysis using the inbuilt algorithms of the ML models also helps us to further evaluate their capability through comparison against the result obtained from other well established techniques.

P18, l17, row?

Thank you, we will correct this to "raw" in the revised version of the manuscript

**References:**

Beven, K.: A manifesto for the equifinality thesis. Journal of Hydrology, 320, 2006.

Choi, H. T. and Beven, K.: Multi-period and multi-criteria model conditioning to reduce prediction uncertainty in an application of TOPMODEL within the GLUE framework, Journal of Hydrology, 332, 2007.

Kirchner, J. W.: Catchments as simple dynamical systems: Catchment characterization, rainfall-runoff modeling, and doing hydrology backward, Water resources research, 45, 2009.

Ratto, M., Castelletti, A., and Pagano, A.: Emulation techniques for the reduction and sensitivity analysis of complex environmental models, Environmental Modelling & Software, 34, 2012.

Stedinger, J. R., Vogel, R. M., Lee, S. U., and Batchelder, R.: Appraisal of the generalized likelihood uncertainty estimation (GLUE) method, Water resources research, 44, 2008.

---

## Editor Comment (EC1) · Dimitri Solomatine (Editor) · 15 Feb 2020

It is an interesting paper, on a topic that deserves attention of the readers. However, referees have correctly pointed out a number of aspect requiring serious attention. One of the referees recommednds "reject" but still, I think the paper can be revised, and would classify the following step as "major revision".

May I suggest to check again comments of Referee 2. I noticed he/she points at papers in HESS (2009 and 2014) where machine learning was used for uncertainty estimation

(MLUE method): neural network is encapsulating results of Monte Carlo uncertainty (GLUE is also MC) analysis and it is used to estimate uncertainty of model predictions for new inputs. In your reply you seem not to notice this suggestion, but it would be advisable to consider doing so.

Please answer all the referess comments, and show how the manuscript is revised according to comments and your answers.

Additionally, it would be perhaps also advisable to look at the papers in WRR and HESS which use machine learning to estimate residual model uncertainty (UNEEC method and its variation) (residual uncertaunty means that it is not Monte Carlo framework that you use).

D.P. Solomatine, D.L. Shrestha. A novel method to estimate model uncertainty using machine learning techniques. Water Resources Res. 45, W00B11, doi:10.1029/2008WR006839, 2009. Wani, O., Beckers, J. V. L., Weerts, A. H., and Solomatine, D. P.: Residual uncertainty estimation using instance-based learning with applications to hydrologic forecasting, Hydrol. Earth Syst. Sci., 21, 4021–4036, https://doi.org/10.5194/hess-21-4021-2017, 2017.

(sorry for point at papers which I co-authored - but you may find that it is quite relevant useful in the context of your research, and to put in the context of the relevant work done earlier, and publihsed in HESS.) I know, the rules say that "Editors themselves should be extra careful in suggesting additional literature." - but in this case I think this advise is justified (especially, for the two papers recommedned by referee 2).

I wish you success in revising the paper.

---

## Author Response (AR1)

Dear editor and reviewers, we are grateful for your thoughtful comments and suggestions. Following is our reply to the points raised in your feedback; and it is structured as comment from reviewer (light blue text) followed by our response to the comment. The specific changes are shown in the marked-up version of the manuscript following the reply to comments section.

**Response to Reviewer #1**

Dear reviewer, as presented in our response to the following general and specific comments; relevant changes have been made and additional explanations and figures have been provided in the revised manuscript.

Reply to the general impression of the reviewer

As you have pointed out under the specific comments (1), the identification of behavioural models through coupling of emulators is affected by multiple factors. It depends on nature of the likelihood measure and its predictability as independent variable (for example in this study, between pLoA and Score). It also depends on the type of fitting model (emulator) used to estimate value of the likelihood measure (in this case the machine learning models).

Although residual-based likelihood measures were used in previous similar studies, as of our best knowledge none of **the emulator based studies** have used pLoA or Score as a response surface, and the limits of acceptability approach in general. And it is for this reason that the first objective of this study was focused on assessing the possibility of using pLoA for the identification of behavioural models using the **coupled** MLMs and the limits of acceptability approach. Further, since the three machine learning models are applied to predict the same response variables followed by the identification of behavioural models using the limits of acceptability approach, the relative performance of RF and KNN (that were not applied in previous studies) can be easily evaluated against the standard ML model, i.e. NNET. And this forms the basis for the second objective of this study, for which the authors believe gives a new insight into the possibility of using RF and KNN as emulators of the MC simulation for application in parameter identification.

To what does one ascribe this conclusion - pLoA or emulation?: "ML emulators and the limits of acceptability approach have performed very well in reproducing the median streamflow prediction both during the calibration and validation periods."

The median streamflow prediction is the result from the **coupled** effect of both the likelihood measure (pLoA) and the specific emulator used to predict the likelihood values.

1. A good emulator (in this case a mapping between $\mathbb{R}^n \to \mathbb{R}$?) may not help to improve the streamflow predictions if the identification metric or the hydrologic models are bad. So the performance of emulation is a somewhat independent question from that of the performance of an identification metric.

This comment is consistent with the response provided above for "the general impression of the reviewer".

From the manuscript, the conclusions suggest that both emulation and pLoA together happen to work well. But even that is doubtful as the paper does not comment on many aspects of emulation.

(a) How do these techniques perform when the models are run fewer number of times, say only 400 times instead of 4000?

The following explanation is provided in the discussion section (Line 20, Page 17) of the revised manuscript

The performance of the coupled MLMs in response to training sample size, however, varies from one MLM to another. For example, RF and KNN did not yield any behavioural model in some of the calibration years when the MLMs are trained with only 400 samples, while NNET has yielded behavioural models in all years. Further, the identified behavioural models using the coupled MLMs with limited sample size had relatively low performance in reproducing the observed streamflow values. For example, NNET, KNN, and RF have respectively yielded an average NSE value of 0.73, 0.70, and 0.65 during the calibration period which is generally lower than the respective values when using the training sample size of 4000. A further assessment of the sample size effect using 2000 training samples have shown only a slight decrease in performance of the identified behavioural models (i.e. a 1-3% decrease in average NSE) as compared to the ones identified using the 4000 samples.

(b) How do these techniques perform with a parameter space of higher dimensionality (n) such that $\mathbb{R}^n \to \mathbb{R}$?)?

Sensitivity of the emulation-based parameter identification to parameter space dimension was not conducted since running the hydrological model used in this study under a distributed setting requires a long time. The model is structured in such a way that, at each time step, the main processes of the model run on each of the grid-cells. This challenge becomes more pronounced when we consider the need for high number of model runs in order to overcome the non-identifiability problem for high parameter space dimensions. Thus, the assessment for effect of parameter space on emulation-based parameter identification might be the subject of our future studies.

(c) Also, what is the added utility of the 95000 simulations in comparison to the already 4000 runs? Any recommendations/comments on the number of samples required for convergence?

The following explanation is provided in Line 9, Page 18 of the revised manuscript

Like most studies based on the GLUE methodology, the main focus of this study was also to get as much behavioural models as possible so as to encapsulate future uncertain conditions. However, only little to no improvement was obtained in most cases when assessed using the available evaluation dataset and the streamflow evaluation metrics used in this study.

(d) How does the emulator perform in extrapolation phase (the 80% calibration, 20% validation separation will not be adequate to show how the emulator may diverge when one uses parameter values away from the training data set. This implication will be more severe when the emulators are used in Bayesian inference and the prior distribution of parameters is not hard-bounded).

As presented in the manuscript (Validation columns in Table 3), capability of the emulators to reproduce the response surface generated directly from the Monte Carlo simulations was further assessed using the 95, 000 samples (S3) in addition to the 20% (test) samples.

(e) And perhaps analysing or commenting on the time efficiency of emulators.

The following text is included in Line 16, Page 11of the revised version of the manuscript:

When it comes to time efficiency of the emulators, they commonly take few seconds to predict the response surface for the 95000 samples as compared to over 24 hours when running the Monte Carlo simulation for a single hydrological year.

2. What new insights do we get from the application of emulation tools to this pLoA metric, apart from the fact that it is a possibility to emulate?

The following explanation is provided in Line 29, Page 21 of the revised version of the manuscript:

The predictability of independent variables varies from one to another. Thus, the application of emulation methods to predict pLoA in this study provides a further insight on the potential and scope of the standard emulator, i.e. NNET and the additional emulators used in this study, i.e. RF and KNN to predict response surfaces other than the residual-based likelihood measures that were applied in previous studies.

"the three MLMs were able to adequately mimic the response surfaces directly estimated from MC simulations". This needs to be made clear (preferably using numbers) in the abstract, discussion and conclusions.

As suggested, we have provided some metric values in the abstract and conclusion sections of the revised version of the manuscript. Following is a text from the abstract section after accommodating the suggestion.

The three MLMs were able to adequately mimic the response surfaces directly estimated from MC simulations with an $R^2$ value of 0.7 to 0.92. Similarly, the models identified using the coupled ML emulators and the limits of acceptability approach have performed very well in reproducing the median streamflow prediction both during the calibration and validation periods with an average Nash-Sutcliffe efficiency value of 0.89 and 0.83, respectively.

3. What is the interpretation of the output generated from behavioral parameters? Do we expect the observations to lie within these bands with a certain frequency? (please refer to Stedinger et al. 2008, for more insights on this debate) If yes, then the reader would like to see reliability (q-q) plots to gauge the performance.

Thank you for your suggestion to the reading material. It provides further insight on uncertainty analysis in hydrological modelling. This theme has been the subject of debate in many hydrology literatures. In order to avoid any confusion with the confidence level expected from the formal Bayesian approach, we have included the following text in Line 19, Page 4 of the revised manuscript:

When using the GLUE methodology, the observations are not expected to lie within the prediction bands at a percentage that equals the given certainty level. However, the modeller can adopt the certainty level specified for producing the prediction limits as a kind of standard for assessing the efficiency of the prediction limits in enveloping the observations (Beven, 2006).

4. How much of the statements made about the efficiency of the emulator are dependent on the choice of the specifications of those machine learning techniques? A paragraph on the meta parameters of this study will be appreciated.

As suggested, the following text and accompanying plots on hyper-parameters of the machine learning models are included in the discussion section of the revised manuscript (Line 25, Page 19).

….. Efficiency of the emulators also depends on their respective hyper-parameter values. Figure 10 shows cross-validation and bootstrap analyses results when estimating the optimal hyper-parameter values of the machine learning models using RMSE for a sample calibration period (year 2011). For NNET (a) two hyper-parameters were optimized using the training dataset, i.e. the weight decay and number of neurons in the hidden layer (hidden units or size). The final values used for this model were a weight decay of 0.001 and hidden units of 10. For KNN (b), the optimal value of nearest neighbours (k) used for the final model was k=10; and for the RF model (c), the optimal number of randomly selected predictors when forming each split (mtry) was 7.

[Figure]

**Figure 10.** Bootstrap and cross-validation based estimates of hyper-parameter values for the three
machine     learning models,  i.e. NNET (a), KNN (b), and RF (c) in a sample calibration period
(year 2011).

5. Some hydrographs will be a useful addition to the existing plots.

As suggested, the following hydrograph plots are included in the revised version of the manuscript
(Figure 4) with subsequent updating of the text in Line 5, Page 13 and the captions of other figures.

[Figure]

**Figure 4**. Simulated and observed streamflow values for the calibration period, i.e. year 2011 (a) and validation periods, i.e. years 2012 (b), 2013 (c), and 2014 (d). The behavioural models are identified using the coupled MLMs (RF, KNN, and NNET) and GLUE pLoA.

6. Please explain why an assumption of 25% for observation error and what will be the effect of choosing a different value on the performance of either GLUE pLoA and the emulation.

In the GLUE LoA methodology, the limits are set with due consideration to the observation and input errors. Since observational error values were not available for the study area, this value was subjectively set based on literature value and observations from a neighbouring catchment plus assumed allowance for input errors. In our previous study, a preliminary assessment on effect of relaxing the limits further, i.e. over 25% while keeping the threshold pLoA at 100% have yielded to the inclusion of non-behavioural models, leading to very low performance during the validation period.

A text explaining this phenomenon is included in the revised version of the manuscript (Line 10, Page 21).

**Response to Reviewer #2**

Dear reviewer, as presented in our response to the following general and specific comments; relevant changes have been made and additional explanations and figures have been provided in the revised manuscript.

This paper presents machine learning methods (MLMs) to emulate MC simulations to identifying behaviour parameter sets of hydrological model. Three MLMs were trained on limited number of MC samples to predict some sort of error or loss function of the MC simulations. Trained models were then used to predict loss function for a large number of samples from which the behavioural parameter sets were identified. While the results look reasonable, there are two main fundamental issues in this manuscript. Authors claimed that the proposed method overcomes computational burden of MC simulations and subjectivity in choosing the likelihood and the threshold value in GLUE. Manuscript fails to provide sufficient evidence to support both claims (see comments below).

I am struggling to find main motivation of this work. It is mentioned that emulators are used to minimize the computational burden of the MC simulation. But this is not completely true. Emulators are used only to predict some sort of likelihood values of the simulation to know whether it should be rejected or not in GLUE framework. Then hydrological models are run with behavioural parameter sets to quantify predictive uncertainty. In other words, MC simulations are still used.

As mentioned in the manuscript, only 5000 MC simulations are run instead of the 95000 from which the behavioural models are identified. The emulators normally take few seconds to predict the response surfaces for the 95000 samples. And this justifies how much the computational cost has reduced as a result of using the MLMs to predict the response surface for the 95000 samples instead of using MC simulations.

This point is clarified in the revised version of the manuscript by including the following text in Line 16, Page 11:

When it comes to time efficiency of the emulators, they commonly take few seconds to predict the response surface for the 95000 samples as compared to over 24 hours when running the Monte Carlo simulation for a single hydrological year.

Indeed, the proposed method does not save computational time when it is required e.g., in real time forecast. For example flood emergency managers want to know the probability of exceeding major flood level at tomorrow noon. There are other ways to emulate MC simulations which are saving computational time in real time application (e.g., Shrestha et al., 2009; Shrestha et al., 2014).

Thank you for bringing the alternative approaches to our attention. The following paragraph is included in the discussion section of the revised manuscript highlighting the general concept and relative time efficiency of the approaches presented in the mentioned reference materials as compared to the equifinality based approaches as used in our study.

In this study, the concept of equifinality was employed for parameter identification and uncertainty analysis, i.e. ensemble of behavioural models were identified with subsequent application for streamflow prediction at different quantile values. In other studies focused on the concept of optimality, machine learning methods were used to directly estimate prediction uncertainty based on MC based uncertainty or historical model residuals from an optimal model. For example, in the MLUE method (Shrestha et al., 2009; Shrestha et al., 2014) MLMs were trained using MC-based uncertainty with subsequent application of the trained MLMs to directly predict model output uncertainty associated with

new input datasets. Similarly, clustering and machine learning techniques were used to estimate the prediction uncertainty associated with a process model through analysis of its residuals during uncertainty estimation based on local errors and clustering (UNEEC) (Solomatine and Shrestha, 2009). In further study, the UNEEC approach was extended in a way that it can explicitly take into account for parametric uncertainty (Pianosi et al., 2010). Wani et al. (2017) have also effectively applied instance-based learning using KNN in order to generate error distributions for predictions of an optimal model. Generally, the UNEEC and its variants are computationally more efficient than those based on the equifinality concept since in the former case only a single model run is required during the forecast period. Uncertainty analysis using emulators coupled to the residual-based GLUE is also expected to entail less computational cost as compared to those coupled with GLUE LoA and its variants.

Another issue in this manuscript is that proposed GLUE pLoA is not convincing. Authors mentioned that the original GLUE has issue in subjectively choosing a likelihood and threshold value for identification of behavioural and non-behavioural parameter sets. They proposed GLUE pLoA to overcome these limitations, however it introduces two additional settings to choose: error bounds and percentage of the model predictions that fall within the error bounds to identify whether given simulation is behavioural and non-behavioural. So proposed method is also subjective, indeed more complex than the original GLUE and requires iterations to choose percentage of the model predictions that fall within the error bounds that satisfy the acceptable CR value.

As mentioned in the original manuscript (Line 13, Page 2; Line 22, Page 4), GLUE pLoA is a time-relaxed variant of GLUE LoA which was introduced in our previous study (Teweldebrhan et al., 2018). Thus, the main goal of this study is to minimize the computational cost when using GLUE pLoA rather than proposing the methodology or comparing against other variants of the GLUE methodology. But we would like to reiterate that it was proposed as part of the endeavour to minimize the rejection of useful models when using the original GLUE LoA formulation rather than to dealing with the subjectivity. Useful models were effectively identified using GLUE pLoA, while all of the 100000 simulations were rejected as non-behavioural models when using the original GLUE LoA formulation (Teweldebrhan et al., 2018).

Verification scores used in this manuscript do not directly test accuracy of emulators to identify behavioural or non-behavioural parameters sets. In this manuscript, RMSE and related measures were used as performance measures of the emulators. However, the problem should be formulated as classification rather than regression if the objective of emulators is to identify whether given simulation is behavioural or non-behavioural.

As indicated in the original manuscript (e.g. Lines 33, Page 3), the emulators were used to predict the response surfaces for new parameter sets. The identification of behavioural models is, however, a result from the **coupled effect** of the emulators in reproducing the response surfaces and the GLUE pLoA in identifying the behavioural parameter sets. Thus, first capability of the emulators to reproduce the response surface was evaluated through comparison of the predicted against MC simulation based values. Then, performance of the behavioural models was evaluated through comparison of their streamflow simulation result against observed values.

We appreciate for the alternative insight you provided us to dealing with the problem. However, in GLUE pLoA, the models are evaluated as ensemble, based on their capability to produce a CR value close to the predefined value, rather than as individual models. For this reason estimating the response

surface using a regression method was found to be more relevant than generating binary values (behavioural/non-behavioural) using classification algorithms.

This term was defined earlier in Line 20, Page 3 of the original manuscript.

P4, L14: What is the basis for 25% as mean observational uncertainty? It is not clear how streamflow limits are computed using this observation uncertainty. Since hydrological model errors are heteroscedastic, applying same value of 25% of the mean observation as error bounds for all time steps would be problematic.

Since no stage-discharge relationship exists for estimating the streamflow uncertainty using the usual practice, i.e. by fitting different rating curves, an assumed value of 25% was adopted based on certain literature values and observational errors analysed for a neighbouring catchment. This value also takes into account incommensurability and uncertainty in the input dataset. The streamflow observational error bounds (limits) of each observation are estimated as ±25% of the corresponding observation, instead of the mean observation. Yet, as the reviewer mentioned since model errors are heteroscedastic mainly in response to the variability in input dataset errors, it would be too strict to expect a given model to satisfy the limits of acceptability criteria in 100% of the observations. And it is this phenomenon that has called the need to introduce the time relaxed variant of the original GLUE LoA formulation (Lines 6-13, Page 21 in the original manuscript).

P4, L27: Define acceptable pLoA. Is it CR from the original GLUE? I wonder what GLUE CR value is. I think this is another subjectivity in this method. Importantly the proposed method relies on original GLUE method to identify acceptable CR. In other words, the proposed GLUE pLoA is not completely independent method, it relies on residual GLUE method to compute its hyper parameters such as acceptable CR.

As indicated in the original manuscript (Line 32, Page 4) the acceptable pLoA is the one that yields a calculated CR value close to the predefined acceptable CR value.

As mentioned in the original manuscript (Line 17, Page 4), the CR value is expressed as the number of observations falling within their respective prediction bounds to the total number of observations (Eq. 1). In this study, the CR value obtained using the residual based GLUE methodology was used for the ease of comparing the result obtained from both methodologies. However, the modeller may also set the acceptable CR value based on previous experience, although this involves some degree of subjectivity. This explanation is included in Line 3, Page 5 of the revised manuscript.

P4, Step 3: "… specified percentage of the total observations." Here is one of subjectivity to identify whether the model simulation is behavioral or non-behavioral. What value is used?

The iteration to get an acceptable pLoA value starts from 100% and decreases further, i.e. relaxed until the desired level of CR is achieved. The reason for relaxing this criterion is provided under the response to the P4, L14 comment. We would, however, like to reiterate that relaxation in the GLUE LoA approach in order to overcome the rejection of useful models is not a new phenomenon. The difference with the previous approaches lies on use of the time relaxed approach than, for example, extending the limits (e.g. Blazkova and Beven, 2009) (Page 2, Line 12; Page 21, Line 8 in the original manuscript).

As suggested, this comment is accommodated in the revised version of the manuscript.

P5, L9: Since all terms of Equation 3 are not defined (e.g. $l$,$u$ ) and assuming $L_e$ in this equation is same as $L_e$ defined in equation 2, I am not sure if the equation is correct. It is not clear whether $e$ is absolute. In either case, for example first expression $\mu_Q = 0, e \leq L_e$ might not be correct. It is better to illustrate Equation (3) with a figure.

Thank you, the notations $l$ and $u$ respectively correspond to $L_e$ and $L_u$. Thus, we have replaced them with the latter notations in order to be consistent with the notations in Equation 2. We have also provided the following illustrative figure accompanying Equation 3 similar to the suggested one. Relevant changes are also made in the reference text.

[Figure]

$$\mu_Q(e) = \begin{cases} 0, e \leq L_e \\ \dfrac{e - L_e}{m - L_e}, L_e < e \leq m \\ \dfrac{U_e - e}{U_e - m}, m < e < U_e \\ 0, e \geq U_e \end{cases}$$

**Figure 1.** A triangular membership function for converting the streamflow prediction error into a normalized criterion.

Here, the notation $e$ is not absolute and thus the expression $\mu_Q = 0, e \leq L_e$ is correct, since a model producing a negative error value of less than the lower observational error bound (which is also a negative value) has 0 degree of membership.

P6, Line 31: 5000 samples may not truly represent the parameter uncertainty. I suggest to use convergence analysis to know the number of samples.

The 5000 samples were used for training and testing of the machine learning emulators. While the behavioural parameter sets that are less than 5000 were identified from the 95000 samples (Section 2.3). The reason for the low number of behavioural samples is partly attributed to the use of uniform parameter distribution and the simple Monte Carlo method for parameter sampling. However, analyses conducted using 50000 and 100000 samples in our previous study have yielded similar parameter and streamflow uncertainty results.

Regarding convergence of the ML training sample size, further analyses were conducted using sample sizes of 400 and 2000; and the following text describing the analyses result was included in the discussion section of the revised manuscript (Line 20, Page 17):

The performance of the coupled MLMs in response to training sample size, however, varies from one MLM to another. For example, RF and KNN did not yield any behavioural model in some of the calibration years when the MLMs are trained with only 400 samples, while NNET has yielded behavioural models in all years. Further, the identified behavioural models using the coupled MLMs with limited sample size had relatively low performance in reproducing the observed streamflow values.

For example, NNET, KNN, and RF have respectively yielded an average NSE value of 0.73, 0.70, and 0.65 during the calibration period which is generally lower than the respective values when using the training sample size of 4000. A further assessment of the sample size effect using 2000 training samples have shown only a slight decrease in performance of the identified behavioural models (i.e. a 1-3% decrease in average NSE) as compared to the ones identified using the 4000 samples.

P11, L5, what is the validation data set? Is it S3?

Here the validation dataset refers to S3 and the corresponding response surface values estimated using the MC simulations. This is clarified in Section 4.1 (Line 7, Page 11) of the revised manuscript.

P13, Table 4: Another widely used cross-validation method is leave out cross-validation. For example, for leave-one-year-out cross-validation, generate simulations in 2011 using model calibrated (e.g., behavioral parameter sets identified) in all data except year 2011, generate simulations in 2012 using model calibrated in all data except year 2012 and so on. Then all simulation data from year 2011, 2012, 2013, and 2014 can be collated to verify the results. This cross validation procedure is expected to produce results that are comparable to those obtainable under operational conditions as the number of data used to fit the model will be similar to that available for operational applications.

Thank you for the suggestion to the alternative cross-validation method. In this study we have preferred to test the model using the worst case scenario, i.e. if we have only one hydrological year for model calibration. Further, as presented in the discussion section, this method allows us to examine the performance of models identified in a given hydrological year when applied under a highly different hydrological condition. A similar approach was used in previous hydrological studies; and it was considered as a more rigorous validation method than the commonly used split-sample methods (e.g. Kirchner, 2009).

P15, Table 5: I strongly suggest replacing Table 5 with distribution plots which is more readable.

Thank you for the suggestion. We have replaced this table with the following box plots displaying the distribution of each parameter under the different emulators.

[Figure]

**Figure 5.** Posterior distribution plots of model parameters identified using the coupled MLMs and MC simulation (RF, KNN, and NNET) as well as those directly identified from the MC simulation (MC)

P15,L3: Section 4.3 is not relevant to this study, so can be deleted.

As discussed in previous studies (e.g. Ratto et al., 2012), sensitivity analysis is often performed in tandem with uncertainty analysis in order to determine which of the input parameters are more important in influencing the uncertainty in the model output. Conducting sensitivity analysis using the inbuilt algorithms of the ML models also helps us to further evaluate their capability through comparison against the result obtained from other well established techniques.

P18, l17, row?
Thank you, this term is changed to "raw" in the revised version of the manuscript

**Response to Editor**

Dear Editor, thank you for your thoughtful comments and suggestions. As presented in our response to the general and specific comments of the reviewers above; we have provided our response to the referee comments. We have also indicated on where the specific changes have been made; and the additional explanations and figures have been provided in the revised manuscript. The specific changes are also shown in the marked-up version following this section.

It is an interesting paper, on a topic that deserves attention of the readers. However, referees have correctly pointed out a number of aspect requiring serious attention. One of the referees recommednds "reject" but still, I think the paper can be revised, and would classify the following step as "major revision".

May I suggest to check again comments of Referee 2. I noticed he/she points at papers in HESS (2009 and 2014) where machine learning was used for uncertainty estimation (MLUE method): neural network is encapsulating results of Monte Carlo uncertainty (GLUE is also MC) analysis and it is used to estimate uncertainty of model predictions for new inputs. In your reply you seem not to notice this suggestion, but it would be advisable to consider doing so. Please answer all the referess comments, and show how the manuscript is revised according to comments and your answers. Additionally, it would be perhaps also advisable to look at the papers in WRR and HESS which use machine learning to estimate residual model uncertainty (UNEEC method and its variation) (residual uncertaunty means that it is not Monte Carlo framework that you use).

D.P. Solomatine, D.L. Shrestha. A novel method to estimate model uncertainty using machine learning techniques. Water Resources Res. 45, W00B11, doi:10.1029/2008WR006839, 2009.

Wani, O., Beckers, J. V. L., Weerts, A. H., and Solomatine, D. P.: Residual uncertainty estimation using instance-based learning with applications to hydrologic forecasting, Hydrol. Earth Syst. Sci., 21, 4021–4036, https://doi.org/10.5194/hess-21-4021-2017, 2017.

(sorry for point at papers which I co-authored - but you may find that it is quite relevant useful in the context of your research, and to put in the context of the relevant work done earlier, and publihsed in HESS.) I know, the rules say that "Editors themselves should be extra careful in suggesting additional literature." - but in this case I think this advise is justified (especially, for the two papers recommedned by referee 2).

Thanks also for bringing the suggestion by Referee #2 and the relevant reference materials to our attention. The following paragraph is included in the discussion section of the revised manuscript highlighting the general concept and merits with regards to time efficiency of the approaches presented in the recommended papers.

[revised manuscript text omitted]

---

## Referee Report (RR1)

REVIEW of the revised paper

**Coupled machine learning and the limits of acceptability approach applied in parameter identification for a distributed hydrological model**

Authors: Aynom T. Teweldebrhan, John F. Burkhart, Thomas V. Schuler, Morten Hjorth-Jensen
Manuscript Number: hess-2019-464

Submitted: **HESSD**

The revised version of the manuscript is a significant improvement over the initial version though I do not agree with some of the responses (see below). The manuscript can be published after minor revision.

I am not fully convinced with the response given by the authors about computational time saved by the emulators. It is not that critical to save computation time for offline simulations. Because calibration or training is generally done one time unless it has to be updated frequently due to significant change in input data distribution. The critical is to save computational time for real time application as mentioned in my comments on earlier version of this manuscript. The proposed method does not provide any benefit over the existing method particularly for real time application. This should be acknowledged at least in the discussion.

Page 4, Line 25: Replace "*The percentage of observations where model predictions fall within the limits*" with "The percentage of the model predictions that falls within the observation error limit"

Page 5, L4: "… *chosen certainty level (e.g. 5-95 %) based on previous experience or literature values.*" Provide references.

Page 3, L 15: prediction error: is this Observation-Simulation or vice versa. It is important to define as error is not absolute (according to response). The response given on page 9 (*Here, the notation e is not absolute and thus the expression $\mu_Q=0, e \leq L_e$ is correct, since a model producing a negative error value of less than the lower observational error bound (which is also a negative value) has 0 degree of membership)p* does not make sense. Let us assume observation Qobs is 100, then according 25% observation error, Le is 75 and Ue = 125. Let corresponding simulation Qsim be 70. According to equation 2, Since Qsim < Le, S(Qsim)=0, this is fine. Now if authors use same notations of Le and Ue in equation 2 and Figure 1 and e = Qsim-Qobs then problem arises for calculating membership of prediction error (See below)
- Case 1: simulation below Le, e.g. Qsim = 70, so e = -30 which is less than Le, so membership = 0
- Case 2, simulation above Le but below m, e.g. Qsim = 80, e = -20, which is also less than Le, so membership = 0
- Case 3, simulation below Ue, but greater than m, e.g. Qsim = 110, e = 10, membership =(125-10)/(125-100)

- Case 4, simulation above Ue, e.g. Qsim = 130, e = 30, membership not 0 because e is not greater than Ue

So notations Le, Ue used in Figure 1 are not same as used in equation 2. In equation 2, notations should be something like this L = Qobs-0.25*Qobs, U = Qobs+0.25*Qobs. Then in Figure 1, it should be Le = L-Qobs and Ue = U-Qobs which will satisfy membership function given in Figure 1. I strongly suggest to use notations of figure of earlier comments in the original version of the manuscript which is also consistent with equation 2.

Table 1: Provide size of S4 on Table 1.

---

## Author Response (AR2)

Dear reviewers, we are grateful for your thoughtful comments and suggestions. Following is our reply to the points raised in your feedback; and it is structured as comment from reviewer followed by our response to the comment (light blue text). The specific changes are shown in the marked-up version of the manuscript following the reply to comments section.

**Response to Reviewer #1**

After going through your reply and revisions, I now change the status of my review to the following: accepted subject to minor revisions.

Thank you again for your thoughtful comments and suggestions. As presented in our response to the following comments, additional explanations have been provided in the revised manuscript.

There is an addition of the following limiting statement in the manuscript on page 4: "When using the GLUE methodology, the observations are not expected to lie within the prediction bands at a percentage that equals the given certainty level."

The additional statement on using the GLUE methodology was mentioned to indicate that a 95% uncertainty level (for e.g.) doesn't necessarily result to bracketing of the observations in 95% of the time. In the past, this topic has been the subject of debate in hydrology literatures. Despite this limitation, the GLUE methodology has been the most widely used uncertainty analysis methodology in hydrology (Stedinger et al., 2008; Xiong et al., 2008; Shen et al., 2012).

Additionally, there is an absence of any uncertainty intervals for the predictions. Therefore, to avoid confusions it necessitates that you mention it explicitly in the abstract as well as in the methods that you are not trying to generate predictive uncertainty estimates using your technique and only use the emulators to identify the most suitable value for the parameters of the model (which finally provide the NSE estimates and the deterministic model predictions in your hydrographs). The whole distribution of parameters from figure 6 is not used in the forward simulations.

In this study, the whole distribution of behavioral parameter sets from Figure 6 was used in forward simulation and the streamflow modelling and prediction uncertainties were quantitatively assessed using the Containing Ratio (CR) index. CR values were computed for each of the calibration and validation years as shown in Table 4. The variability in the level of uncertainty between the calibration and validation years; and between the coupled machine learning models (within each year) were also discussed in section 4.2. The relevant statement in the methodology section (Page 4, L18) is also rephrased in the revised version of the manuscript for a better clarification.

**Response to Reviewer #2**

The revised version of the manuscript is a significant improvement over the initial version though I do not agree with some of the responses (see below). The manuscript can be published after minor revision.

Thank you again for your thoughtful comments and suggestions. As presented in our response to the following comments; relevant changes have been made and additional explanations have been provided in the revised manuscript.

I am not fully convinced with the response given by the authors about computational time saved by the emulators. It is not that critical to save computation time for offline simulations. Because calibration or training is generally done one time unless it has to be updated frequently due to significant change in input data distribution. The critical is to save computational time for real time application as mentioned in my comments on earlier version of this manuscript. The proposed method does not provide any

benefit over the existing method particularly for real time application. This should be acknowledged at least in the discussion.

In the past, emulators have been used to save computational costs both during model calibration and prediction. As mentioned in the manuscript (e.g. Page 3, L11) "The main goal of this study is to emulate the time consuming MC simulation **for parameter identification** through coupling of the machine learning models with the time relaxed limits of acceptability approach." Similarly, the value of other methodologies focused on reducing the computational cost during the prediction period such as the MLUE and UNEEC based approaches are also discussed in the manuscript (Page 20, L7). However, on whether calibration time or prediction time is critical may, among other factors, depend on the modelling purpose and length of weather forecasts. For example, in some sectors of operational hydrology the identified behavioral models (which are much less in number than the MC simulations) are used to generate predictions for the coming few days (hours) for applications in short term planning. In other cases, width of the prediction window might be limited by the availability of reliable forecasted weather data for longer periods. Whereas, during the calibration period it is usually required to run thousands of Monte Carlo simulations over historical data from several years which sometimes may take from days to months.

Page 4, Line 25: Replace "*The percentage of observations where model predictions fall within the limits*" with "The percentage of the model predictions that falls within the observation error limit"

The suggested statement, i.e. "The percentage of the model predictions that falls within the observation error limit" refers to the measure of streamflow prediction uncertainty, i.e. the containing ratio index (CR), while "*The percentage of observations where model predictions fall within the limits*" refers to the pLoA metric.

Page 5, L4: "… *chosen certainty level (e.g. 5-95 %) based on previous experience or literature values.*" Provide references.

Reference was provided in the revised version of the manuscript

Page 3, L 15: prediction error: is this Observation-Simulation or vice versa. It is important to define as error is not absolute (according to response).

This was defined in the revised version of the manuscript as ($Q_{obs}$ - $Q_{sim}$) (Page 5, L17)

notations Le, Ue used in Figure 1 are not same as used in equation 2. In equation 2, notations should be something like this L = Qobs-0.25*Qobs, U = Qobs+0.25*Qobs. Then in Figure 1, it should be Le = L-Qobs and Ue = U-Qobs which will satisfy membership function given in Figure 1. I strongly suggest to use notations of figure of earlier comments in the original version of the manuscript which is also consistent with equation 2.

Although in both expressions $Le$ and $Ue$ represent the acceptable error bounds, the underlying expressions are not the same. In Equation 2 they are calculated as the observed value plus or minus of the observational error, i.e. $Le$ = Qobs-0.25*Qobs and $Ue$= Qobs+0.25*Qobs, while in Figure 1 the deviations from perfect match (i.e, zero error) are calculated as $Le$ = m-0.25*Qobs and $Ue$= m+0.25*Qobs. Thus, in order to avoid ambiguity, the notations for upper and lower observational error bounds in Equation 2 are respectively changed to $LQe,i$ and $UQe,i$ with additional clarifications provided in the revised manuscript. However, using the notations from the figure of earlier comment would be less consistent to the expressions used in the analysis. The suggested figure assumes Qsim as

direct input to the membership function $\mu Q$, i.e. $\mu Q(Q\text{sim})$, whereas the expression in Figure 1 takes the prediction error e= $Qobs - Q\text{sim}$ as its input, i.e. $\mu Q(e)$. This can also be noticed from the figure ordinate labeled as 'prediction error'. Thus, using the notations from the figure of earlier comment would be inconsistent with the underlying expression used in the analysis. This inconsistency was also reflected in the following sample calculation by the reviewer.

The response given on page 9 *(Here, the notation e is not absolute and thus the expression $\mu_Q = 0, e \leq L_e$ is correct, since a model producing a negative error value of less than the lower observational error bound (which is also a negative value) has 0 degree of membership)* does not make sense. Let us assume observation Qobs is 100, then according 25% observation error, Le is 75 and Ue = 125. Let corresponding simulation Qsim be 70. According to equation 2, Since Qsim < Le, S(Qsim)=0, this is fine. Now if authors use same notations of Le and Ue in equation 2 and Figure 1 and e = Qsim-Qobs then problem arises for calculating membership of prediction error (See below)

- Case 1: simulation below Le, e.g. Qsim = 70, so e = -30 which is less than Le, so membership = 0
- Case 2, simulation above Le but below m, e.g. Qsim = 80, e = -20, which is also less than Le, so membership = 0
- Case 3, simulation below Ue, but greater than m, e.g. Qsim = 110, e = 10, membership =(125-10)/(125-100)
- Case 4, simulation above Ue, e.g. Qsim = 130, e = 30, membership not 0 because e is not greater than Ue

The ambiguity reflected in this sample membership function calculation seems to partly stem from the notations used in Equation 2 and Figure 1 which is now fixed in the revised version of the manuscript (reply to previous comment). Further, the sample calculations by the reviewer seem to assume $Q\text{sim}$ instead of $e$ as ordinate values of $\mu Q(e)$. The following table demonstrates the results obtained from $\mu Q(e)$ when using the right input to the expression, i.e. $e = Qobs - Q\text{sim}$.

Given:

| | | Error bounds based on m±25%$Qobs$ error | |
|---|---|---|---|
| $Qobs$ | $m$ | $Le$ | $Ue$ |
| 100 | 0 | -25 | 25 |

Solution for the cases when using the expression in Figure 1:

| $Q\text{sim}$ | e | Case | $\mu Q(e)$ |
|---|---|---|---|
| 70 | 30 | $e \geq Ue$ | 0 |
| 80 | 20 | $m < e < Ue$ | 0.2 |
| 100 | 0 | $e = m$ | 1 |
| 110 | -10 | $Le < e \leq m$ | 0.6 |
| 130 | -30 | $e \leq Le$ | 0 |

Table 1: Provide size of S4 on Table 1.

The size of S4 obtained under each calibration year and the coupled machine learning models (MLMs) is provided in the revised version of the manuscript. Since S4 is an output from the analysis whose value varies between calibration years and between MLMs, it is presented in Table 4 with an accompanying brief discussion provided in Section 4.2.

**References:**

[revised manuscript text omitted]